# A streamlined pipeline for multiplexed quantitative site-specific N-glycoproteomics

Pan Fang[1,6], Yanlong Ji [1,2,3,6], Ivan Silbern [1,4], Carmen Doebele[2,5], Momchil Ninov [1,4], Christof Lenz [1,4], Thomas Oellerich[2,3,5], Kuan-Ting Pan [2,3✉] & Henning Urlaub [1,4✉]

Regulation of protein N-glycosylation is essential in human cells. However, large-scale, accurate, and site-specific quantification of glycosylation is still technically challenging. We here introduce SugarQuant, an integrated mass spectrometry-based pipeline comprising protein aggregation capture (PAC)-based sample preparation, multi-notch MS3 acquisition (Glyco-SPS-MS3) and a data-processing tool (GlycoBinder) that enables confident identification and quantification of intact glycopeptides in complex biological samples. PAC significantly reduces sample-handling time without compromising sensitivity. Glyco-SPS-MS3 combines high-resolution MS2 and MS3 scans, resulting in enhanced reporter signals of isobaric mass tags, improved detection of N-glycopeptide fragments, and lowered interference in multiplexed quantification. GlycoBinder enables streamlined processing of Glyco-SPS-MS3 data, followed by a two-step database search, which increases the identification rates of glycopeptides by 22% compared with conventional strategies. We apply SugarQuant to identify and quantify more than 5,000 unique glycoforms in Burkitt's lymphoma cells, and determine site-specific glycosylation changes that occurred upon inhibition of fucosylation at high confidence.

[1] Bioanalytical Mass Spectrometry Group, Max Planck Institute for Biophysical Chemistry, 37077 Göttingen, Germany. [2] Hematology/Oncology, Department of Medicine II, Johann Wolfgang Goethe University, 60590 Frankfurt am Main, Germany. [3] Frankfurt Cancer Institute, Goethe University, 60596 Frankfurt am Main, Germany. [4] Institute of Clinical Chemistry, University Medical Center Göttingen, 37075 Göttingen, Germany. [5] German Cancer Consortium/ German Cancer Research Center, 69117 Heidelberg, Germany. [6]These authors contributed equally: Pan Fang, Yanlong Ji. ✉email: kuan-ting.pan@kgu.de; henning.urlaub@mpibpc.mpg.de

With an increasing understanding of protein *N*-glycosylation and its site-specific regulation in physiology and pathology[1,2], a pressing need is arising for large-scale mass spectrometry (MS)-based identification and quantification of intact *N*-glycopeptides. Conventional analytical approaches often release the glycan moiety chemically or enzymatically from proteins before performing MS analysis of either the resulting oligosaccharides or the de-glycosylated peptides[3], thus deliberately discarding information about site specificity. Instead, analysis of intact *N*-glycopeptides (i.e., peptides that derive from endoproteolytic digestion of glycosylated proteins, but still carry the intact glycan moieties) is essential to determine the linkage between protein and glycan, and to profile the microheterogeneity of glycosylation at each site[4,5]. Significant improvements have been made with regard to methods for intact glycopeptide characterization, including glycopeptide enrichment[6–8], optimized MS acquisition strategies[8], advanced fragmentation techniques[9,10], and database search algorithms[11]. These have helped address technical difficulties resulting from heterogeneous and complex glycopeptide structures and their less informative fragmentation behavior in tandem mass spectrometry (MS2). Nevertheless, these developments have largely focused on increasing glycopeptide identification rates rather than on reliable, global glycopeptide quantification.

Both iTRAQ and TMT isobaric labeling have been used for large-scale quantitative glycoproteomic studies[12,13]. Despite the advantages of isobaric labeling in quantitative intact glycoproteomics (Supplementary Note 1), standard MS2-based methods for the analysis of labeled glycopeptides often suffer from reduced identification rates and impaired accuracy of quantification[14]. The recently developed synchronous precursor selection (SPS)-MS3 approach has improved quantitative accuracy in isobaric mass tag labeling experiments by reducing co-selected precursor interference at the cost of scan speed[15,16], but its application to glycoproteomics has not yet been explored. Quantification of intact glycopeptides on a large scale remains technically challenging, and software tools for quantitative data processing are still lacking.

Here we introduce SugarQuant, an integrated workflow for large-scale, global glycopeptide identification and quantification (Fig. 1). SugarQuant comprises (i) lysis of cells in the presence of SDS, (ii) protein extraction, (iii) protein concentration and endoproteolytic digestion using protein aggregation capture (PAC)[17,18], (iv) multiplex TMT labeling, (v) *N*-glycopeptide enrichment by zwitterionic HILIC (ZIC-HILIC) followed by (vi) basic reverse phase (bRP) prefractionation, and (vii) LC-MS[3] analysis using Glyco-SPS-MS3, which generates high-resolution MS2 and MS3 product ion scans. Data processing using the GlycoBinder tool (viii) combines MS2 and MS3 fragment ions for *N*-glycopeptide identification, and extracts TMT reporter-ion intensities from MS3 scans for each identified *N*-glycopeptide-spectrum-match (GPSM). Glyco-Binder consolidates redundant GPSMs with their quantitative values, and reports multi-dimensional quantification results for unique glycoforms, glycosylation sites, and glycan compositions in an accessible table-based format. As a proof of concept, we apply SugarQuant to quantitatively map the protein glycosylation patterns in Burkitt's lymphoma cells treated with varying doses of 2-deoxy-2-fluoro-L-fucose (2FF), a guanosine diphosphate fucose analog that inhibits cellular fucosylation. We demonstrate the capabilities of SugarQuant for site-specific determination of reduced N-fucosylation following 2FF treatment, which is missed by conventional MS2 analysis. Our findings uncover 2FF-sensitive *N*-glycosylation sites and reveal 2FF-mediated changes in *N*-glycosylation on key effectors in B-cell receptor-mediated signaling.

## Results

### PAC enables fast preparation of TMT-labeled glycopeptides.
We developed a PAC-based workflow for fast preparation of TMT-labeled *N*-glycopeptides (Fig. 1). Using human lymphoma cell lysates (DG75 and Daudi), we evaluated and optimized cell lysis conditions, PAC bead types, ZIC-HILIC enrichment, and bRP chromatography to achieve a shorter handling time and good recovery of *N*-glycopeptides (Supplementary Note 2). Cell lysis in the presence of 4% SDS recovered 9% more unique proteins and 20% more unique glycoforms than the use of 1% RapiGest, a MS-compatible cleavable detergent (Supplementary Fig. 1a). PAC allowed efficient removal of SDS with reduced sample preparation time and led to 13.8% and 26% more identified glycoforms and glycosites compared to conventional acetone precipitation, respectively (Supplementary Fig. 1b). In particular, additional prefractionation of ZIC-HILIC enriched *N*-glycopeptides by bRP chromatography before LC-MS analysis resulted in the identification of >1.5 times more *N*-glycopeptides when compared to repeated direct LC-MS injections of the same sample amount obtained after ZIC-HILIC, even when using prolonged chromatographic gradients for the latter (Supplementary Fig. 1c). Importantly, bRP chromatography offers high resolution, as evidenced by 94.1% of glycopeptides which were exclusively identified in at most two fractions (Supplementary Fig. 1d) whereas almost one-third of glycopeptides were identified in five replicates when using replicate injections, and only 26.7% of glycopeptides were unique to a single run (Supplementary Fig. 1e). bRP prefractionation is thus suited to maximize the productivity of LC-MS acquisition time.

We then introduced TMT-labeling into the workflow to allow for highly multiplexed relative quantitation. In its most recent implementation, TMT enables up to 16-fold multiplexing, which greatly increases sample throughput and allows for sophisticated experimental setups[19]. We confirmed that ZIC-HILIC enriches TMT-labeled *N*-glycopeptides, either from purified human IgM or from whole cell lysate, with comparable efficiency and specificity as their unlabeled counterparts (Supplementary Fig. 1f). The optimized workflow (Supplementary Fig. 1g) including cell lysis, PAC-based clean-up and proteolysis, TMT-labeling, ZIC-HILIC enrichment, and off-line prefractionation of *N*-glycosylated peptides prior to LC-MS, enables one-day sample preparation for large-scale multiplex quantitative glycoproteomics.

### Glyco-SPS-MS3 for improved identification and quantification of TMT-labeled *N*-glycopeptides.
Continuous developments of MS methods in past decades[10,20] have greatly improved the characterization of glycopeptides, but revealed shortcomings in the quantitation of their TMT-labeled counterparts. We therefore evaluated both MS2- and MS3-based approaches available on Orbitrap Tribid mass spectrometers with regard to their performance for the identification and quantitation of glycopeptides.

We used higher-energy collisional dissociation (HCD) on both unlabeled and labeled *N*-glycopeptides obtained from human serum IgM (Supplementary Fig. 2) to determine the influence of collision energy on the information content of the MS/MS spectra of TMT-labeled glycopeptides. On a purely identification-based level, the highest number of labeled glycopeptides was identified at 30–35% NCE (Normalized Collision Energy), as compared to an optimum value of 25–30% for non-labeled glycopeptides (Supplementary Fig. 2). We next evaluated the production of different structural moiety-specific fragment types (i.e. fragments related to peptide sequence and to glycan composition, and TMT reporter ions) at different NCE settings (Fig. 2 and Supplementary Fig. 3). We note that the most intense glycan product ions are observed at NCE <35%, peptide fragment ions at NCE

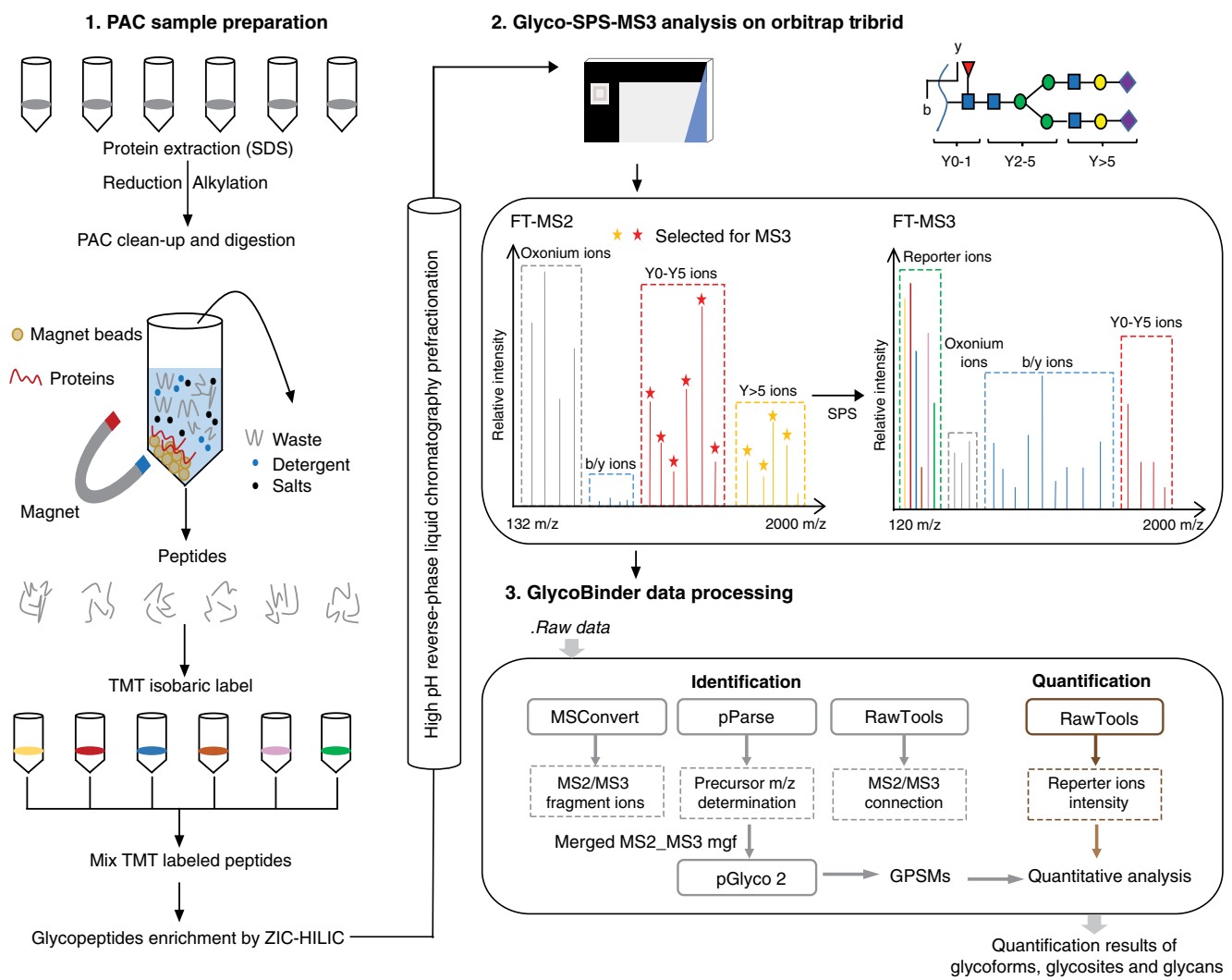

**Fig. 1 Scheme of SugarQuant for multiplexed, quantitative and site-specific glycoproteomics.** The workflow of SugarQuant includes PAC-based method for easy preparation of TMT-labeled glycopeptides, a Glyco-SPS-MS3 acquisition method for confident identification and accurate quantification of intact glycopeptides, and the GlycoBinder script for automated data processing.

~40–45%, and TMT reporter ions at NCE >45% (Fig. 2a). Such a wide range of NCEs for the various types of fragment ions poses a challenge for achieving both identification and quantification of N-glycopeptides when using HCD-MS2 only. Our observations corroborate earlier published datasets[12,13] that showed near one-order less intense reporter ions for labeled glycopeptides compared to labeled linear peptides in HCD-MS2, even with NCE as high as 45% (Supplementary Fig. 4). Possibly the breakage of the more labile glycosidic bonds absorbs most of the collisional energy and thus limits the generation of reporter ions[14]. Low reporter ion intensities will compromise TMT-based quantification sensitivity and reproducibility. In addition, MS2-based methods suffer from impaired quantitative accuracy due to interference caused by co-isolation[21].

Based on previous MS3 strategies for glycopeptide characterization[22,23] and the recently developed SPS-MS3 method[15,16], we generated an MS acquisition workflow for the analysis of TMT-labeled N-glycopeptides, namely Glyco-SPS-MS3 (Fig. 2b and Supplementary Note 3). It applies HCD at low-NCE (~25%) to produce intense glycan Y-series ions carrying the intact peptide backbone. The 10 most abundant fragment ions in the range of 700–2,000 m/z are then co-selected and co-fragmented with higher NCE (35–40%) HCD in MS3 to generate abundant peptide

b-/y-ions as well as TMT reporter ions (Supplementary Fig. 5). The dominant glycan Y ions observed in glycopeptide MS2 scans inherently bear TMT tags, precluding the possibility of selecting an untagged peptide y-ion for SPS-MS3 quantification as frequently happens when analyzing regular tryptic peptides without lysine residues. Besides, glycan Y ions often produce better peptide backbone fragmentation than the corresponding intact glycopeptides, which improves overall peptide sequence coverage. Glyco-SPS-MS3 reconciles the divergent collision energy requirements of different fragment ion types, and results in increased quantitative accuracy by removing MS2-co-isolated interferences. As an adjustment to the originally proposed SPS-MS3 workflow, we chose to detect both MS2 and MS3 fragment ions with high resolution and high accuracy in the Orbitrap, and use HCD in MS2 as well as MS3 (Fig. 2b and Supplementary Fig. 6). This allows subsequent combination of MS2 and MS3 fragment ions derived from the same N-glycopeptide precursor in post-acquisition processing in a straightforward manner (see below). Synchronous selection of multiple MS2 fragments for MS3 analysis further boosted the overall sensitivity (Supplementary Fig. 7).

A shortcoming of any MS3-based method is the longer duty cycle. However, our results showed that increased automatic gain

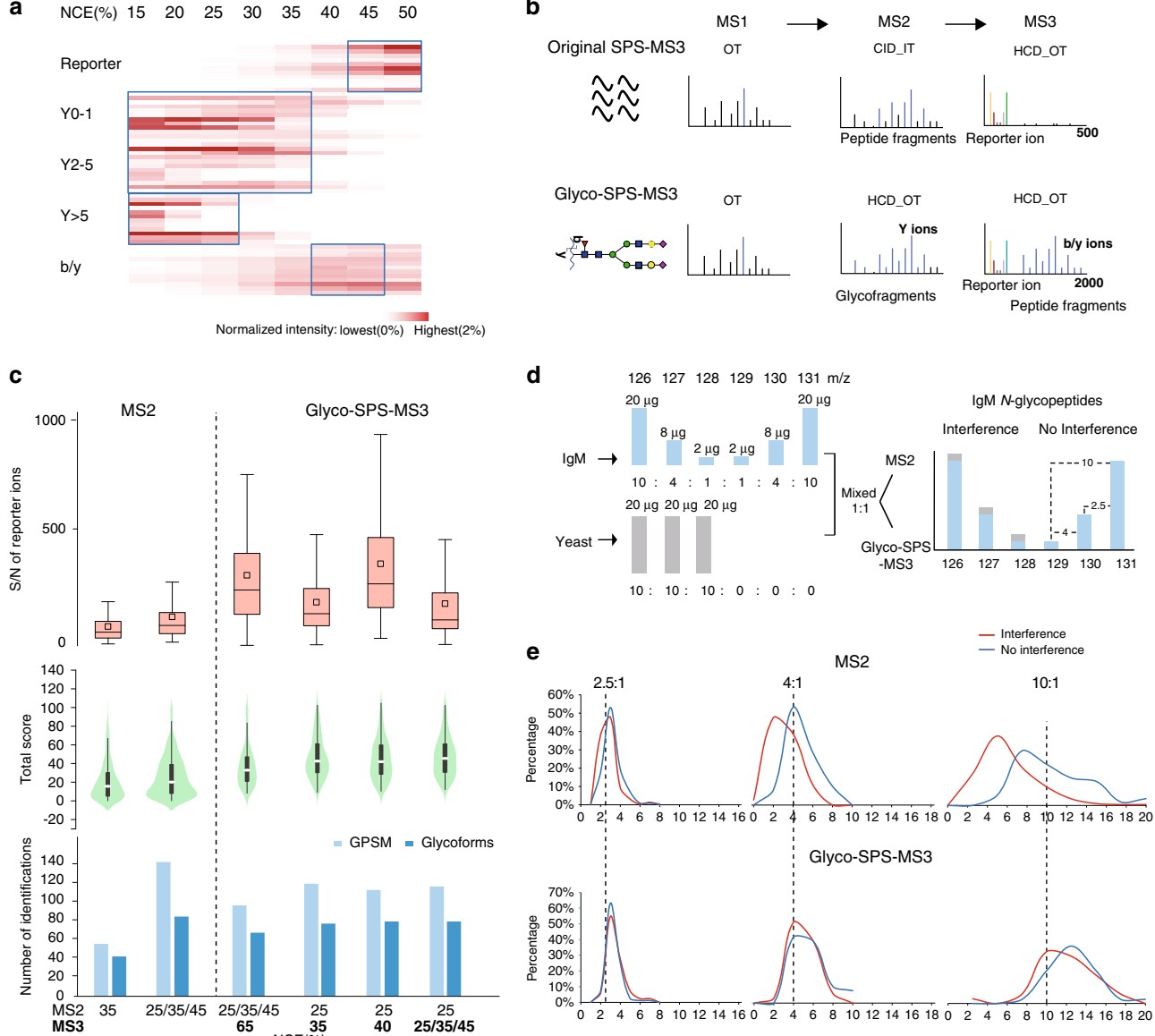

**Fig. 2 Glyco-SPS-MS3 method for confident glycopeptides identification and accurate quantification. a** A heatmap showing normalized intensities of different product ion types (see Supplementary Fig. 3 for nomenclature) of 12 TMT-labeled IgM glycopeptides detected in MS2 experiments using various NCEs. Optimal NCEs for individual product ion types were highlighted with blue boxes. **b** Schematic workflows of the original SPS-MS3 as compared with Glyco-SPS-MS3. Ions being selected for MS2 or MS3 fragmentation are in blue. TMT reporter ions are in red. MS detectors and fragmentation modes used in the methods are shown (IT for ion trap and OT for Orbitrap). **c** Comparisons of the distributions of TMT-reporter S/N ratios (upper panel, $n = 116, 306, 181, 226, 206,$ and $218$), pGlyco total score of GPSMs (middle, $n = 336, 504, 188, 233, 219,$ and $227$) and the number of identifications (bottom) obtained using MS2 or Glyco-SPS-MS3 methods with varied NCE settings. $n = 2$ independent measurements of TMT-labeled IgM glycopeptides. Boxplots show the median (centerline), mean (squares), first and third quartiles (box edges) and 1.5× the interquartile range (whiskers). Outliers are not shown. **d** Preparation of the IgM-yeast mixture and anticipated outcome of interfered glycopeptide quantification due to precursor co-isolation. **e** Determined reporter ion ratios of IgM glycopeptides from the IgM-yeast mixture using MS2 or Glyco-SPS-MS3 methods. Channels with and without yeast interference are shown in blue and red, respectively. Predicted ratios are indicated. Source data are provided as a Source Data file.

control (AGC) target values and prolonged ion injection time (IT) benefited *N*-glycopeptide identification in a standard MS2 analysis in Orbitrap Tribid mass spectrometers (Supplementary Fig. 8a). Those settings improve *N*-glycopeptide identification but significantly slowdown the overall scan speed in a MS2 analysis. Accordingly, the AGC and IT in Glyco-SPS-MS3 were adjusted (Supplementary Fig. 8b–e) in order to maintain a cycle time of ~3 s, comparable with MS2 without compromising the identification of *N*-glycopeptides. We note that Glyco-SPS-MS3 triggered 30% fewer precursors than MS2, but reached up to 3.6 fold GPSM

identification rates depending on the NCEs used (Supplementary Fig. 9).

Next, we evaluated various single or stepped NCE (sNCE) settings in the Glyco-SPS-MS3 versus standard MS2 methods to optimize both the signal-to-noise (S/N) ratios of the TMT reporter ions as well as the numbers of identified GPSMs and glycoforms. In addition, peptide and glycan subscore distributions were analyzed to better understand the effect of NCE settings (Fig. 2c and Supplementary Fig. 10). While MS2 with sNCE moderately outperformed Glyco-SPS-MS3 (Fig. 2c, bottom

panel) in identification, Glyco-SPS-MS3 significantly improved the overall matched scores in a database search, particularly of glycan scores (Fig. 2c, middle panel and Supplementary Fig. 10a, b). Furthermore, TMT reporter ion intensities in Glyco-SPS-MS3 were boosted by a median factor of up to 400%. Among other NCE settings (Supplementary Fig. 10c, d), we deemed the combination of 25–30% NCE for MS2 and 35–40% for MS3 as the best combination to achieve both high identification rates and high TMT reporter ion intensities. The application of sNCE in either MS2 or MS3 within Glyco-SPS-MS3, such as the stepping 25–25/35/45, did not result in any improvement in the sequencing of subsequent database search of glycopeptides (Fig. 2c).

To evaluate the impact of our Glyco-SPS-MS3 approach on co-isolation interference of the TMT reporter ions and therefore quantitative accuracy, we prepared a 6plex TMT-labeled IgM-yeast peptide mixture. We labeled IgM peptides separately with individual TMT reagents (126–131) and pooled them afterwards in ratios of 10:4:1:1:4:10, respectively. Yeast peptides were labeled with only the first three channels of TMT reagents (126, 127, and 128) and mixed in a 1:1:1 ratio. The labeled IgM peptides were added to the labeled yeast peptides in a 1:1 (wt/wt) ratio of total peptide amount (Fig. 2d). The mixture was analyzed by either Glyco-SPS-MS3 or by a standard MS2 method. Ideally, we should be able to accurately quantify accurately the IgM $N$-glycopeptides with the predefined ratios of 10:4:1:1:4:10 against the labeled yeast peptide background. Upon co-isolation, however, the labeled (co-selected) yeast peptides contribute to extra amounts of TMT reporter ions that can distort the ratios in the channels with which they interfere (i.e., 126, 127, and 128) while leaving the remaining ones (129, 130, and 131) unchanged. Indeed, we accurately recovered the predefined 2.5-, 4-, and 10-fold changes, respectively, in the non-interfered channels, both by MS2 and by Glyco-SPS-MS3 (Fig. 2e). In contrast, for the interfered channels (126, 127, and 128), the standard MS2 method resulted in skewed median fold changes of 2.2, 2.0, and 4.2, while our Glyco-SPS-MS3 was still able to detect TMT reporter ratios more accurately, i.e., 2.9, 3.8, and 11.2 (Fig. 2e). Our results demonstrate that Glyco-SPS-MS3 allows accurate quantification because of the increased TMT-reporter ion intensity and the minimized interference arising from co-isolation.

**GlycoBinder for single-step quantitative $N$-glycoproteomics data-processing.** There is an increasing number of open software tools which use specialized scoring algorithms designed to achieve reliable glycopeptide identification[20]. However, none of these supports database search of MS3 spectra and multiplexed quantification with isobaric mass tags. To make full use of the benefits of Glyco-SPS-MS3 for LC-MS-based identification and quantification of labeled $N$-glycopeptides, we developed GlycoBinder, an R-based tool. It combines existing computational tools to automatically extract and merge MS2 and MS3 fragments obtained on the same precursor ion into pseudo-fragment ion spectra, and to conduct glycopeptide identification by database search (Fig. 3a and Supplementary Note 4). Specifically, GlycoBinder uses MSconvert[24] to convert MS2 and MS3 spectra into mgf format, and further uses RawTools[25] to generate a table listing the respective scan numbers of all MS3 scans and their parent MS2 scans. GlycoBinder then merges all MS2 and MS3 fragment ions accordingly using a predefined mass tolerance (1 ppm by default). GlycoBinder executes pParse[26] to re-assign the mono-isotopic peak of each precursor. Subsequently, GlycoBinder searches the merged pseudo-fragment spectra with corrected precursor $m/z$-values for $N$-glycopeptide identification using the algorithm pGlyco 2[27], which supports command-line execution

and FDR control on both peptide and glycan levels. At the end, GlycoBinder utilizes RawTools again to extract TMT reporter intensities from raw MS files and then supplements each GPSM with corresponding quantitative values.

We evaluated the performance of GlycoBinder for data analysis of TMT-labeled N-linked glycopeptides derived from DG75 cells in a human database search (in total 48 Glyco-SPS-MS3 raw files). GlycoBinder with the use of pParse identified 6% more GPSMs than without pParse. The Thermo Proteome Discoverer (PD) platform, which enables similar MS2-MS3 spectra merging (see Methods section), identifies 9% or 15% less GPSMs when compared with GlycoBinder with or without pParse, respectively (Fig. 3b). In terms of quantification of multiplexed $N$-glycopeptides, we confirmed that the reporter intensities extracted by RawTools are consistent with the values reported by PD (Fig. 3c).

We next investigated whether $N$-glycopeptide identification by GlycoBinder can be improved by searching against more specialized databases. We used a reviewed human proteome database from Swiss-Prot ("Human-reviewed", 20,303 entries, March 2018), a curated glycoprotein database from Swiss-Prot (Glycoprotein-reviewed, 4824 entries, March 2018), an in-house "B-cell-Specific glycoprotein" database with 974 glycoproteins that were repetitively identified in our inventory studies, and finally a non-related database consisting of a random selection of 1000 proteins (Random-1000) from the Human-reviewed database (Methods, Supplementary Data 1). GlycoBinder identified similar numbers of unique glycoforms from the Human-reviewed and the Glycoprotein-reviewed databases (4381 vs. 4335, respectively) at 2% FDR with 80% glycoform overlap. (Fig. 3d and Supplementary Fig. 11a). Despite the fact that the B-cell-specific database contains only 974 proteins—a 95% decrease compared to the Human-reviewed—GlycoBinder achieved 10% more identifications on this project-specific database than on the Glycoprotein-reviewed database. The search against the Random-1000 database resulted in the identification of only 243 of unique glycoforms, suggesting that using a smaller database does not spontaneously cause overfitting of the database search. The specificity of the database rather than its size is crucial for the improvement of identification[28].

A highly specific, experimentally based database carries the disadvantage that only glycosylation sites on known glycoproteins will be identified and quantified. Therefore, we implemented a two-step database search, in which we first search the raw data against the entire human protein database (here: Human-reviewed). Identified glycoproteins are subsequently used for a second database search of the same raw data (Fig. 3a). With this approach, we aimed to rescue low-stoichiometry glycoforms from the glycoproteins identified in the first search. Indeed, the two-step search boosted the identifications to 5367 unique glycoforms and 855 glycosites, representing respectively a 22.5% and an 8.5% increase as compared with the single search using the Human-reviewed database. More than 90% of the glycosites were identified in both the first and second searches (Supplementary Fig. 11b). In addition, our results indicated that the two-step search helped to identify more lower-scored glycoforms that were missed in the first search, at a fixed 2% FDR (Supplementary Fig. 11c).

In summary, GlycoBinder not only supports standard database searches with user-defined FASTA protein sequence databases, but also allows an automated two-step search for glycopeptide identification with considerably higher sensitivity. In addition, GlycoBinder uses the annotated result from pGlyco 2 software and propagates the corresponding GPSMs to different levels of quantification, including unique glycosylation sites, unique glyco-forms, and unique glycan compositions (Fig. 4a and Supplementary Note 4). The propagation of quantitative values of GPSMs is

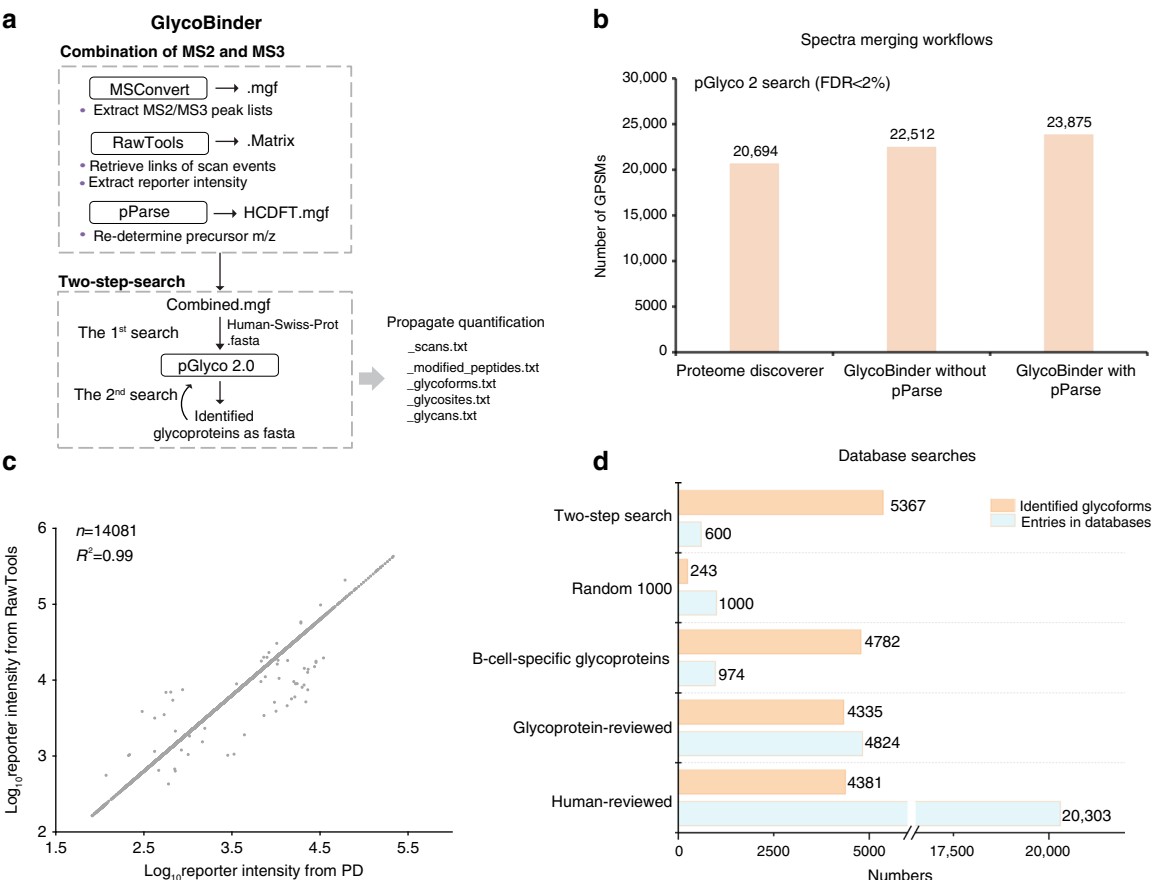

**Fig. 3 GlycoBinder enables streamlined MS3 data handling for advanced intact glycopeptide identification and accurate multiplexed quantification.**
**a** The schematic workflow of GlycoBinder. **b** Different spectra-merging workflows produced different GPSM identification rates. TMT-labeled *N*-glycopeptides derived from DG75 cells were analyzed by Glyco-SPS-MS3 (48 raw files). All combined mgf files were searched using pGlyco 2 against a human database under 2% FDR. **c** Correlation of reporter ion intensities extracted by RawTools and PD. Each circle represents the extracted reporter ion intensity from a scan. A total of 14,081 reporter ions were compared. **d** Comparison of the two-step search with single searches using various databases (see main text). Sizes of the databases (light blue) and the resulting numbers of identified glycoforms (light brown) are shown. Source data are provided as a Source Data file.

performed in the same manner as in quantitative proteomics, where peptide quantifications are propagated to the protein level[29]. To maximize the utility's value, GlycoBinder also supports raw files acquired with common MS2 methods for those who have no access to the MS3-capable instruments. GlycoBinder is available on GitHub (https://github.com/IvanSilbern/GlycoBinder).

**Multiplexed quantification of *N*-glycoproteome on fucosylation-inhibited human Burkitt's lymphoma cells.** To test the suitability of the SugarQuant workflow for large-scale glycosylation analysis, we applied it to the analysis of global fucosylation changes in human Burkitt's lymphoma cells under the influence of 2-deoxy-2-fluoro-L-fucose (2FF). Fucosylated glycans play vital roles in a variety of biological and pathological processes, and aberrant fucosylation is associated with human diseases, such as the increase of core fucosylation in cancers[30]. De-fucosylation using fucose analogs like 2FF suppresses cell proliferation and migration in human liver cancer cells, suggesting a possible way to retard tumor development[31]. However, it remains elusive whether 2FF has any effect on the glycoprotein quantities per se, which might in turn affect the protein site-specific glycosylation in (cancer) cells[32]. We applied SugarQuant to the quantitative analysis of glycopeptides derived from 2FF-treated human Burkitt's lymphoma cells (DG75) in order to gain a deeper understanding of the underlying mechanism of

2FF-treatment and discover 2FF-sensitive glycosylation sites (Fig. 4b). DG75 cells were treated with 2FF at different concentrations (60–600 μM) for 3 days. The overall decrease in protein fucosylation was confirmed by western blotting against a biotinylated fucose-specific lectin (*Aleuria Aurantia* lectin, AAL, Supplementary Fig. 12a). Proteins were extracted and digested using the PAC method, and the resulting peptides were labeled with TMT6plex. Labeled glycopeptides were enriched with ZIC-HILIC, and further separated by bRP and analyzed by LC-MS using the Glyco-SPS-MS3 or the MS2 method. Databases were searched and *N*-glycopeptides were annotated by using GlycoBinder.

We found that the standard MS2 method (Supplementary Table 2) identified slightly more glycoforms (6%), glycosites (3%), and glycans (2%) than the Glyco-SPS-MS3 method (Supplementary Fig. 12b). In contrast to the 13–30% of decreased identification rates commonly observed in proteomics or phosphoproteomics by switching from MS2 methods to SPS-MS3[15,33], SugarQuant reached a comparable level of sensitivity in glycoproteomics as compared to MS2, in spite of the slower acquisition rate. Importantly though, Glyco-SPS-MS3 outperformed the MS2 method to precisely determine the decrease of TMT-labeled glycoforms under 2FF treatment at all concentrations (Fig. 4c and Supplementary Fig. 13). Our results showed good reproducibility (Pearson correlation $r = 0.59$–0.98) among

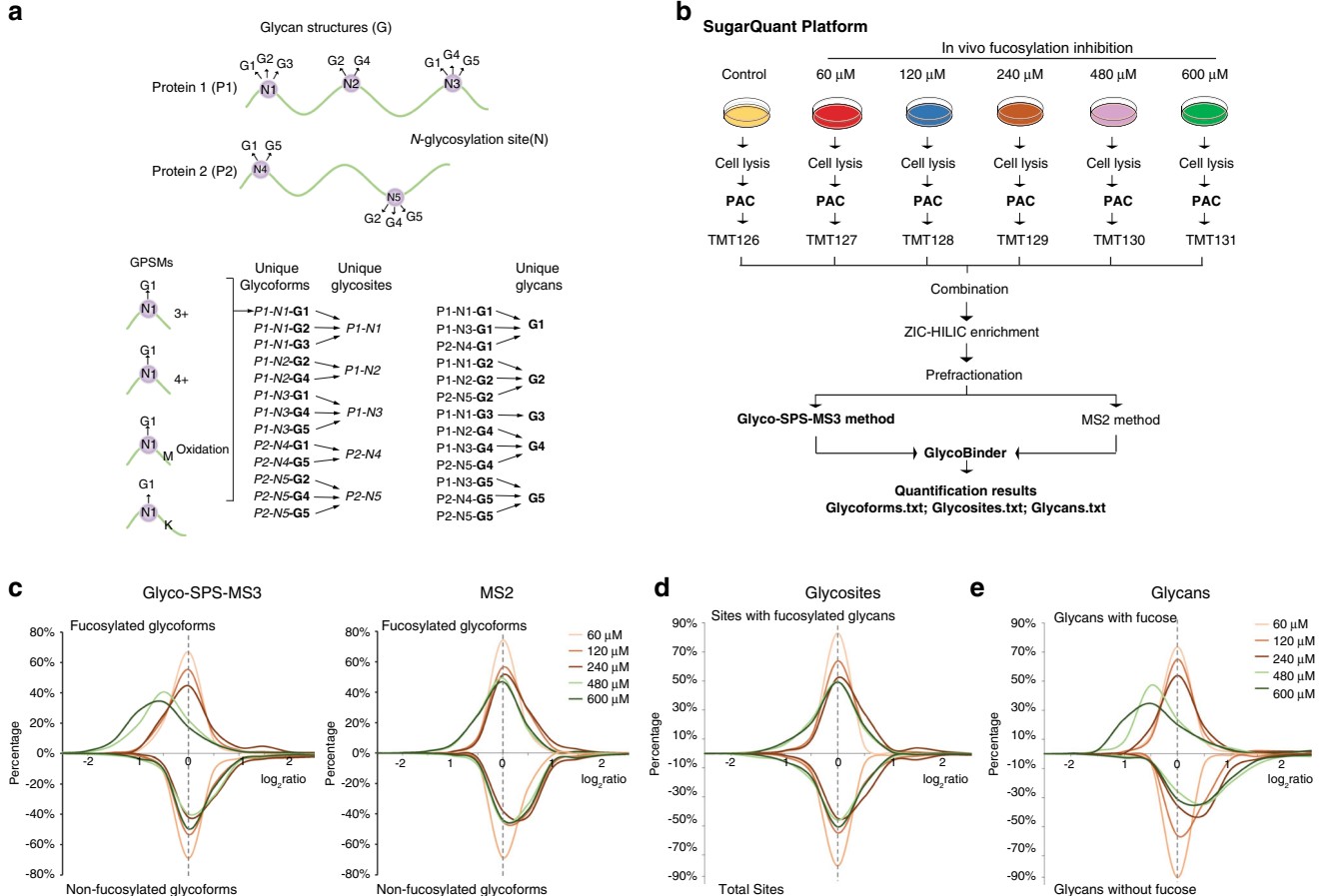

**Fig. 4 Multi-dimension quantitative *N*-glycoproteomics of 2FF-treated human Burkitt's lymphoma cells. a** Schematic explanation of the propagation of quantitative information from GPSMs to unique glycoforms, glycosites, and glycans in this study. For one *N*-glycosylation site (N) on a protein (P), multiple glycans (G) may exist on the site, resulting in multiple glycoforms (represented as P-N-G). Quantification of unique glycoforms are achieved by summing TMT-reporter ion intensities of all involving GPSMs resulted from miscleavages, modifications, and different charge states. Quantified glycoforms are further combined for glycosite (P-N) quantification. As for a unique glycan, the quantification values of all glycoforms with this particular glycan in the sample are combined. **b** The SugarQuant workflow conducted for the 2FF treatment on DG75 cells. *n* = 2 biologically independent cells. **c** Aligned ratio distributions of glycoforms quantified via the Glyco-SPS-MS3 (left) or the MS2 method (right) upon 2FF treatment. Different 2FF concentrations were color coded as shown in the figure. For direct comparison, we separate fucosylated glycoforms (upper panel) from non-fucosylated ones (lower panel). **d**, **e** Similar to **b**, but showing the 2FF-induced changes at the levels of glycosites (left) or glycan composition (right) determined by the Glyco-SPS-MS3. Source data are provided as a Source Data file.

biological and technical replicates (Supplementary Fig. 14). Since only 40% of the identified *N*-glycopeptides contain a fucose moiety (Supplementary Data 2), we assume that the more accurate quantification with Glyco-SPS-MS3 is largely caused by reduced co-isolation interference of TMT reported ions, whereas in the MS2 method the non-fucosylated glycopeptides particularly hamper the reliable quantification of fucosylated ones. Again, Glyco-SPS-MS3 showed significantly enhanced TMT reporter ion intensities in this large-scale study (Supplementary Fig. 12c).

In this *N*-glycoproteomics analysis of human lymphoma B cells, we identified 5367 unique glycoforms, which contained 414 glycan compositions on 855 glycosites from 528 glycoproteins (Supplementary Data 2–4). Our results revealed substantial microheterogeneity of the site-specific *N*-glycosylation in B cells. We identified at least five different glycan compositions on more than 27% of all glycosites, with an average of 6.3 glycoforms per *N*-glycosylation site. In its extreme, four identified glycosites (on Slc3a (424), IGHM (46), Lamp1 (249), and PTPRC (380)) harbor more than 100 different glycoforms (Supplementary Fig. 15a). The majority of the glycoforms bore glycans containing 8–14

monosaccharide moieties with molecular masses of 2–3 kDa (Supplementary Fig. 15b, c). Of the 528 glycoproteins identified, however, only 13% have three or more different glycosites (Supplementary Fig. 15d).

Using the propagation function implemented in GlycoBinder, we extended the TMT-based quantification to the glycosite- and glycan-centric levels (Fig. 4d, e). Our results show a substantial reduction in numbers of fucosylated glycans under 2FF treatment, concomitant with a 2FF concentration-dependent increase of non-fucosylated glycans (Fig. 4e). At 480 μM 2FF, the quantities of 321 glycoforms on 138 glycosites within 105 proteins changed significantly (Z score > 2 or < −2) and 48 glycosites from 46 glycoproteins are under-glycosylated under 2FF treatment, and 42 of these glycosites contain significantly changed glycoforms (Supplementary Data 3 and 4). Annotation of glycoproteins shows that the 2FF-affected glycoproteins are tightly associated with the description of lymphoma and enriched for putative drug targets in various cancers (Fig. 5a and Supplementary Data 5). The majority of those lymphoma-related proteins are enzymes or cell surface molecules with highly confident functional connections between them, including CD molecules, integrins, and

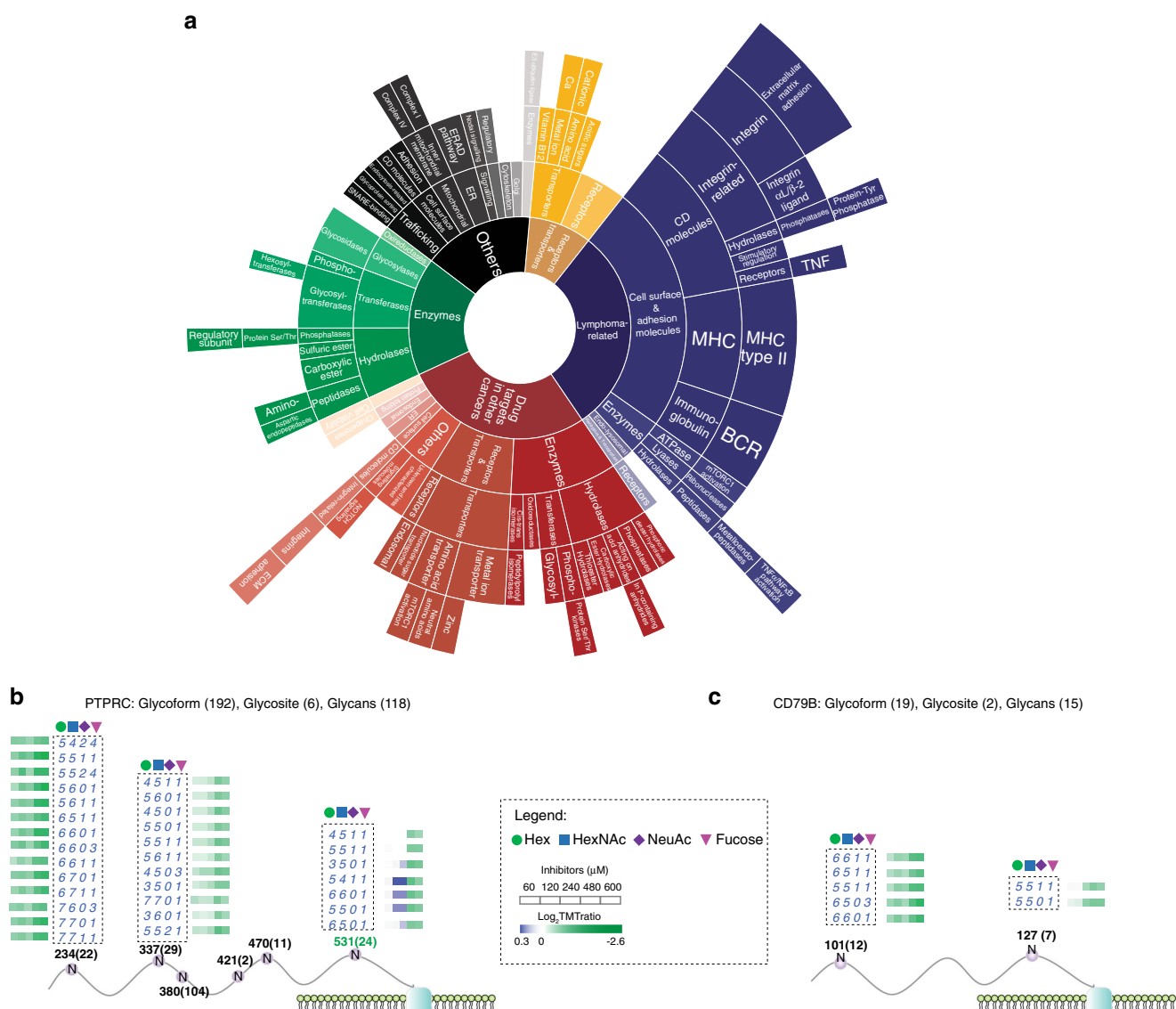

**Fig. 5 Functional annotation of 2FF-affected glycoproteins. a** A sunburst plot visualizing the functional classification of the glycoproteins that bear significantly changed glycoforms upon 2FF treatment. Significant regulation is defined using Z score >2 or <−2. **b**, **c** Visualization of all quantified glycosylation sites (N) on PTPRC (**b**) and CD79b (**c**) under 2FF treatment with specified residue number and number of quantified glycoforms in bracket. A glycosite that showed a significantly changed glycosylation level is marked in green. Regulated glycoforms are shown with their glycan compositions presented as the numbers of Hex, HexNAc, NeuAc, and Fucose (dashed box). Determined fold changes of glycoforms upon 2FF treatment with various concentrations are shown by color-coded rectangles. Source data are provided in supplementary data 5 and Source Data file.

MHC class II molecules (Supplementary Fig. 16). For example, the fucosylated glycan epitope, Lewis^x, is expressed on Hodgkin's Reed–Sternberg cells (often carried by CD98 and ICAM-1, both are 2FF-affected glycoproteins) and is an established diagnostic marker for patients with Hodgkin's lymphoma[34,35].

The glycoproteins affected by 2FF treatment include major players in B-cell receptor (BCR)-mediated signaling. For instance, in the human receptor-type tyrosine-protein phosphatase C (PTPRC), which is a tyrosine phosphatase involved in the regulation of B-lymphocyte activation, SugarQuant identifies and quantifies in total 192 glycoforms on 6 glycosites (Fig. 5b). While fucosylated glycoforms are significantly reduced on Asn-234, 337, and 531, only the glycosite Asn-531 exhibits decreased glycosylation occupancy (given that no 2FF-induced proteome change was detected (Supplementary Data 6)). Also, 19 glycoforms of two glycosites on CD79B, which is the B-cell antigen receptor

complex-associated protein beta chain for initiation of the signal transduction cascade activated by the B-cell antigen receptor complex, were also quantified by SugarQuant (Fig. 5c). Both quantified CD79B glycosites contain 2FF-induced decreases of fucosylated glycoforms, but none of them shows significantly changed glycosylation occupancy. Our results suggest that 2FF site-specifically regulates glycosylation of human Burkitt's lymphoma cells on important BCR effectors, which may in turn influence downstream BCR signaling.

## Discussion

We present here a developed SugarQuant pipeline for multiplexed quantification of site-specific N-glycoproteomics. SugarQuant utilizes a PAC-based method to ensure sufficient solubilization of membrane-associated glycoproteins, complete and fast removal of

detergents, efficient TMT-labeling, and selective glycopeptide enrichment with reduced handling time and sample loss. PAC-based sample preparation has been used in a substantial range of proteomics experiments that extends from in-depth analyses of cell lines to clinical samples[17,18]. It assures the applicability of Sugar-Quant to a range of biological systems. For MS acquisition, SugarQuant makes use of the optimized Glyco-SPS-MS3 to enhance the detection of both glycan and peptide fragments for more confident glycopeptide identification. Glyco-SPS-MS3 also significantly improves quantitative accuracy by lowered co-isolation interference and increased TMT-reporter ion intensities. Finally, SugarQuant includes GlycoBinder, an R-based tool to seamlessly combine MS2 and MS3 scans of Glyco-SPS-MS3 data, perform database search, and propagate GPSMs for glycoform-, glycosite-, and glycan-centric quantification. We successfully applied the SugarQuant to determine site-specific glycosylation changes in fucosylation-inhibited DG75 cells. Our results not only represent the present largest gly-coproteomics data of human lymphoma cells but also demonstrate the capability of SugarQuant for accurate multi-dimension quantification.

## Methods

**Cell culture and fucosylation inhibition.** DG75 and Daudi cells were cultured in RPMI medium (Invitrogen) supplemented with 10–20% (vol/vol) heat-inactivated fetal bovine serum (Invitrogen), penicillin/streptomycin (Invitrogen), and L-glu-tamine (Invitrogen) at 37 °C with 5% $CO_2$. DG75 cells (DSMZ no.: ACC 83) were provided by Prof. Dr. Thorsten Zenz at the National Centre for Tumor Diseases (NCT) in Heidelberg, Germany. Daudi cells (DSMZ no.: ACC 78) were purchased from Leibniz Institute DSMZ-German Collection of Microorganisms and Cell Cultures GmbH in Braunschweig, Germany. Cells were grown to ~90% confluency before harvesting and provided as cell pellets aliquoted at ~$1 × 10^7$ cells per tube.

For fucosylation inhibition, cells with an initial concentration of $0.2 × 10^6$ cells per mL in medium were treated with various concentrations of 2-fluoro-L-fucose (2FF) (Carbosynth Limited) in dimethyl sulfoxide (DMSO) The final concentrations of 2FF in medium were 60 μM, 120 μM, 240 μM, 480 μM, and 600 μM, respectively. The final DMSO concentration was kept at 0.1% (vol/vol) in all cases. After incubation for 72 h, cell pellets were collected after washing three times with cold PBS, then stored at −80 °C until use.

**Preparation of peptides using conventional approaches.** Cell pellets were lysed with either one of the four buffers: Buffer 1 consisted of 4% SDS (w/v), 50 mM HEPES, pH 8.0; Buffer 2 consisted of 8 M urea in 50 mM Triethylammonium bicarbonate (TEAB); Buffer 3 consisted of 0.1 % (w/v) RapiGest SF surfactant (Waters) in 50 mM TEAB; and Buffer 4 consisted of 1% (w/v) RapiGest in 50 mM TEAB. All the four lysis buffers were supplemented with 1× Halt Protease and Phosphatase Inhibitor Cocktail (Thermo Scientific). All the samples were sonicated for 15 min (15 s on, 15 s off) using Bioruptor at 4 °C. After centrifugation at 14,000 × g for 15 min, the supernatants were collected and the protein con-centration were determined using Pierce BCA Protein Assay Kit (Thermo Scien-tific). All the proteins were reduced and alkylated with 10 mM tris(2-carboxyethyl) phosphine hydrochloride (TCEP, 500 mM stock, Sigma)) and 20 mM iodoaceta-mide at 37 °C for 60 min in the dark. For samples in buffer 1, 3 times sample volume of cold (−20 °C) acetone was added, and mixed samples were stored at −20 °C for at least 3 h or overnight to precipitate the proteins. The samples were then centrifuged for 10 min at 14,000 × g, and the supernatants were removed carefully without disturbing the protein pellets. The protein pellets were resolu-bilized in 0.1% Rapigest in 50 mM TEAB by thorough pipetting or vortexing. The proteins were digested using trypsin at an enzyme to protein ratio of 1:50 at 37 °C overnight. For samples in buffer 2, we diluted the samples 8-fold with 50 mM ammonia bicarbonate (ABC). After overnight trypsin digestion, the resulted pep-tides were desalted by Oasis HLB columns (Waters). For samples in buffer 3, trypsin were directly added for overnight digestion. For samples in buffer 4, we did the same as the samples in buffer 3, except that we diluted 10-fold with 50 mM TEAB before digestion.

**PAC-based preparation of TMT-labeled peptides from cells.** Cell pellets were lysed in buffer 1. Cell protein extraction, reduction, and alkylation were the same as mentioned above. We compared three different types of magnetic beads, including Sera-Mag SpeedBeads with a hydrophilic surface (GE Healthcare, cat.no. 45152101010250, Magnetic Carboxylate Modified), Sera-Mag SpeedBeads with a hydrophobic surface (GE Healthcare, cat.no. 65152105050250, Magnetic Carbox-ylate Modified) and MagReSyn HILIC (ReSynBio, cat.no. MR-HLC005) with a mixed-mode functional surface. The magnetic beads were rinsed twice with water on a magnetic rack prior to use. The beads were added to protein lysates at the optimal working ratio of 10:1 (wt/wt, beads to proteins). The required minimum

bead concentration is 0.5 μg/μL in order to provide sufficient surface for the immobilization of aggregated proteins. We then added acetonitrile to protein lysates to a final percentage of 70% (vol/vol). The samples were allowed to stay off the rack for 10 min at room temperature (RT), followed by resting on the magnetic rack for 2 min at RT. The supernatant were discarded and the beads were then washed for three times with 80% (vol/vol) ethanol. Beads were resuspended in 50 mM TEAB containing sequencing grade modified trypsin (1:50 of enzyme to protein concentration) and incubated at 37 °C for 4 h or overnight in a Thermo-Mixer with mixing at 800 rpm. After digestion, we placed the tubes on a magnetic rack for a few minutes and transferred the supernatant to a fresh tube. TMT labeling procedure (Thermo Scientific) was performed according to the manu-facturer's instruction.

**Glycopeptide enrichment.** All the digested samples were acidified by adding 10% (vol/vol) trifluoroacetic acid (TFA) to a final concentration of 1% followed by centrifugation at 14,000 × g for 20 min. The supernatant were then dried in a SpeedVac concentrator. We re-dissolved the dried peptides in loading buffer consisting of 80% (vol/vol) acetonitrile (ACN) and 1% TFA and maintained the peptide concentration at around 4 mg/ml. Meanwhile, we weighted out the ZIC-HILIC beads (5 μm, Welch) according to the peptide to bead ratio of 1:50 (wt/wt), and washed three times using the loading buffer. We loaded all beads onto a yellow pipette tip pre-packed with coffee filter. Samples were loaded five times (at least 4 min each time) followed by three times wash with loading buffer. The retained glycopeptides were eluted with 100 μL 0.1% TFA twice. The collected eluates were dried in a SpeedVac concentrator.

**Basic reverse phase fractionation.** Basic reverse phase analysis was performed on an Agilent 1100 series HPLC system. Enriched glycopeptides were dissolved in 50 μL mobile phase A (10 mM ammonium hydroxide in water, pH 10). Elution was performed at a flow rate of 60 μL/min using mobile phase A and B (10 mM ammonium hydroxide in 80% acetonitrile, pH 10) with a Waters XBridge C18 column (3.5 μm particles, 1.0 mm inner diameter, and 150 mm in length). The gradient was 2% B for 5 min, to 34% B in 37 min, to 60% B in 8 min, to 90% B in 1 min, held at 90% B for 5.5 min, to 2% B in 0.5 min, and then held at 2% B for 7 min (64 min total runtime). Peptides were detected at 214 nm and 58 fractions were collected along with the LC-separation in a time-based mode from 6 to 64 min. Fractions were then pooled into eight concatenated fractions.

**Preparation of TMT-labeled IgM peptides.** Human IgM purified from human serum (Sigma, I8260) was reconstituted at the final concentration of 1 mg/ml in 0.05 M Tris-HCl, 0.2 M sodium chloride, 15 mM sodium azide, pH 8.0. The pro-teins were heated at 95 °C for 10 min. After cooling to RT, protein reduction, alkylation and digestion were the same as mentioned above. The digested samples were acidified by adding TFA to a final concentration of 1% (vol/vol). The insoluble particles in the samples were removed by centrifuging at 14,000 × g for 10 min using a benchtop centrifuge. After cleaning up with Oasis HBL columns, the resulting peptides were dried in a SpeedVac concentrator. TMT labeling procedure (Thermo Scientific) was performed according to the manufacturer's instruction

**IgM and yeast peptide interference model.** Yeast protein prepared from S. cerevisiae cells were purchased from Promega (Cat. V7341). Yeast proteins in 6.5 M urea/50 mM Tris-HCl (pH 8) at a protein concentration of 10 mg/ml were thawed on ice. Protein reduction and alkylation were the same as mentioned above. Dilute the urea to 1 M using 50 mM Tris-HCl (pH 8). Trypsin (Promega) was added at a trypsin:protein ratio of 1:50 at 37 °C. After overnight incubation, the samples were acidified by adding TFA to a final concentration of 1% (vol/vol). The insoluble particles in the samples were removed by centrifuging at 14,000 × g for 10 min using a benchtop centrifuge. The samples were cleaned up with Oasis HBL columns. The resulted peptides were dried in a SpeedVac concentrator.

We separately labeled IgM digests with individual TMT6 reagents and pooled them together afterward with the ratio of 10:4:1:1:4:10. In contrast, yeast peptides were labeled with only the first three channels of TMT6 reagents (126, 127, and 128) and mixed equally (Fig. 2d). We then spiked the pooled IgM peptides into the yeast peptide mixture with an equal amount. The mixed samples were then cleaned up with Oasis HBL columns followed by analysis with either Glyo-SPS-MS3 or standard MS2 methods (see below).

**Lectin blotting.** DG75 cell pellets with and without 2FF treatment were lysed in 4% SDS, 0.1 M Tris-HCL, pH 8.0. Protein concentrations were determined using BCA. For each condition, 20 μg proteins were reconstituted in 1× NuPAGE LDS Sample Buffer (Invitrogen) and separated on 4–12% NuPAGE Novex Bis-Tris Minigels (Invitrogen). The proteins separated in gel were transferred onto a nitrocellulose membrane (GE Healthcare). The membrane was blocked with 5% (w/v) bovine serum albumin (Sigma) in TBST overnight at 4 °C. The membrane was then incubated with biotinylated lectins AAL (vector laboratories; 3 μg/mL in blocking buffer) for 1 h at room temperature. The membrane was washed three times using TBST and incubated with Strep-Tactin®-HRP conjugate (iba) in TBST for 1 h. After washing, the signal was visualized using a chemiluminescence detection system (ECL, GE Healthcare) and detected on X-ray film.

**LC-MS analysis**. TMT-labeled or unlabeled glycopeptides were resuspended in 5% ACN, 0.1% FA, and subjected for LC-MS/MS analysis using Orbitrap Fusion Tribrid or Lumos Mass Spectrometers (Thermo Scientific), both coupled to a Dionex UltiMate 3000 UHPLC system (Thermo Scientific). IgM glycopeptides were firstly concentrated on a C18 trap column (3 cm long; inner diameter, 100 μm; outer diameter, 360 μm) for 3 min and then separated on a home-made analytical column (ReproSil-Pur 120 C18-AQ, 1.9 μm pore size, 75 μm inner diameter, Dr. Maisch GmbH, 30 cm or 50 cm (only for Supplementary Fig. 1)) using an one-hour gradient at a flow rate of 300 nl/min. Mobile phase A and B were 0.1% (vol/vol) formic acid (FA) and 80% ACN, 0.08% FA, respectively. The gradient started at 5% B at 3 min, increased to 50% B in 42 min, and then to 70% B in 4 min. After washing with 90% B for 6 min, the column was re-equilibrated with 5% B. For glycopeptides from DG75 cells, no trap column was used. After sample loading with 2% B in 49 min, the gradient increased to 5% B in 4 min, continued to 38% B in 147 min, and then to 60% B in 16 min, followed by the above-mentioned washing and re-equilibration steps. For proteomics analysis, a trap column was used. The gradient started at 8% B at 3 min, increased to 40% B in 90 min, and then to 60% B in 13 min, followed by the above-mentioned washing and re-equilibration steps.

MS parameters evaluated in Glyco-SPS-MS3 were listed in Supplementary Table 1. The instrument settings optimized for Glyco-SPS-MS3 and those used for common MS2 acquisition were summarized in Supplementary Table 2. For proteomics analysis on Orbitrap Fusion, the following settings were used. MS1 settings: Detector Type-Orbitrap, Orbitrap Resolution-120 k, Scan Range-350–1,550, Maximum injection time-50 ms, AGC target-4e[5], RF Lens-60%, and Data Type-Profile. MS2 settings: Isolation mode-Quadrupole, Isolation window-1.6 $m/z$, Scan range mode-Auto normal, First mass-110, Activation type-HCD, Collision energy (%)−40, Detector type-Orbitrap, Orbitrap resolution-60 K, Maximum injection time-60 ms, AGC target-5e[4], and Data type-Profile.

For the evaluation of IgM $N$-glycopeptide HCD fragmentation, each selected precursor in MS1 survey scans was subject to eight consecutive MS2 scans with NCE15%, 20%, 25%, 30%, 35%, 40%, 45%, and 50%, respectively (Fig. 2a and Supplementary Figs. 2 and 3). The intensities of product ions detected under different NCEs were extracted and normalized to the total ion current of the respective spectra. For the evaluation of detector type and fragmentation type in MS2 methods, each selected precursor from IgM samples was subject to 4 sequential MS2 scans, including Orbitrap_HCD, Ion trap_HCD, Orbitrap_CID and Ion trap_CID (Supplementary Fig. 5).

**LC-MS data processing**. For intact glycopeptide identification and quantification, raw files were processed via GlycoBinder (Supplementary Note 4), which is available on GitHub (https://github.com/IvanSilbern/GlycoBinder). Parameters used for pGlyco 2 include fully specific trypsin digestion with maximal two missed cleavage and mass tolerance for precursors and fragment ions of 10 and 20 ppm, respectively. We considered cysteine carbamidomethylation and TMT0 (or TMT6) on peptide N-termini and lysine residues as fixed modifications and methionine oxidation as a variable modification. Please refer to the GitHub page for other default settings of the GlycoBinder. The reviewed human protein database (Human-reviewed) was downloaded from Swiss-Prot (March 2018, human, 20,303 entries). In addition, the "Glycoprotein-reviewed" (4824 entries, March 2018) database was downloaded from Uniprot using the keyword of "glycoprotein". The database "Random-1000" was generated by random selection of 1000 protein sequences from the "Human-reviewed" database using an R base function $sample()$. The "B-cell-specific" database was built using the glycoproteins identified from ~250 runs for DG75 or Daudi cells in our lab.

For the identification of IgM glycopeptides, we included only IGHM_HUMAN and IGJ_HUMAN in the FASTA file, and only GPSMs with PepScore >7 and GlyScore >8 reported by pGlyco 2 were used for following analysis. For DG75 samples, we used the total FDR <2% for both the first and second database search. The reported "TotalScore", "PepScore", "GlyScore", "GlyIonRatio" and "PepIonRatio" by pGlyco 2 were used for the evaluation of identification confidence[27]. PepScore and GlyScore are the scores for peptide sequence based on b/y ion and for glycan composition based on Y ions, respectively. TotalScore is the scores for intact glycopeptide based on the weighted sum of the PepScore and GlyScore according to the pGlyco soring algorithm. GlyIonRatio and PepIonRatio are the ratios of #matched Y ions/#theoretical Y ions and #matched peptide ions/#theoretical peptide ions, respectively.

For the evaluation of detector types and fragmentation types in MS2 methods (Supplementary Fig. 6), Proteome discoverer (PD, version 2.2) was used to separate MS2 spectra according to their respective settings and then convert to mgf. Each of the resulting mgf files was searched against IGHM_HUMAN/IGJ_HUMAN database using Byonic (version 2.8.2) separately. Byonic parameters included 10 p.p.m. of precursor ion tolerance and 20 ppm of fragment ion tolerance for Orbitrap and 0.5 Da of fragment ion tolerance for Iontrap, full trypsin specificity on both termini, up to two missed cleavages, cysteine carbamidomethylation (+57.02146 Da) as a static modification, and methionine oxidation (+15.99492 Da) as a variable modification.

For the IgM-Yeast interference model, the raw MS files were processed using GlycoBinder and searched against a protein database including sequences from

IGHM_HUMAN and IGJ_HUMAN followed by sequences of proteins encoded by all known S.cerevisiae ORFs. FDR < 2% for both the first and second database search was used.

For processing Glyco-SPS-MS3 results using PD, the nodes of "Spectrum selector" and "Spectrum grouper" were used for converting and combining spectra from MS2 and MS3, respectively, with the following settings. Precursor mass criterion: same singly charged mass; Precursor mass tolerance: 0.1 ppm; Max. RT difference: 0.04 min; Allow mass analyzer mismatch: False; Allow MS order mismatch: True. The resulting mgf files were used for pGlyco 2 searches. Alternatively, the mgf files can be used for glycopeptide identification and quantification using the Byonic node in the PD platform.

Proteomics results were processed via PD with built-in Sequest HT and Percolator nodes using the following settings: the "Human-reviewed" database, fully specific trypsin digestion, maximum two missed cleavages, mass tolerance for precursors and fragment ions of 10 and 20 ppm, respectively, cysteine carbamidomethylation and TMT6 on peptide N-termini and lysine residues as fixed modifications, methionine oxidation as a variable modification, and target FDR as 1%.

**Data analysis**. GlycoBinder reports directly the integrated TMT-reporter ratios of unique glycoforms, glycosites, and glycan compositions in separate text files. To account for quantitative errors introduced before TMT-labeling, TMT-ratios of glycoforms determined in each glycoproteomics experiment on DG75 cells were normalized by the median TMT-ratio of respective proteomics analysis. Significant regulation is defined using $Z$ score >2 or <−2. Unless mentioned elsewise, all figures were made using Excel 2016.

For the sunburst plot, we investigated and annotated the protein functions of 2FF-affected glycoproteins manually via surveying literature relevant to the respective gene and protein names (including alternative names) and keywords of "cancer", "lymphoma" or "B cell" (Supplementary Data 5). We summarized the information about the biological function(s) and subcellular localization of all proteins and classified them accordingly into multiple categories. For example, the first inner layer contains the groups of lymphoma-related, drug target in other cancer, enzymes and others. The number of proteins in each category is proportional to the size of the corresponding categorical area in the sunburst plot. The interaction networks of proteins were done by STRING (version 11.0)[36].

Boxplot, violin plot, and density plot were created using OriginPro 2020 (9.7.0.185). In the boxplots, centerlines and squares in plotted boxes indicate the median and mean, respectively. The upper and lower ends of box shows the 75th and 25th percentiles. The extreme lie shows 1.5× the interquartile range.

**Reporting summary**. Further information on research design is available in the Nature Research Reporting Summary linked to this article.

## Data availability

The mass spectrometry proteomics data have been deposited to the ProteomeXchange Consortium via the PRIDE[37] partner repository with the dataset identifier PXD018349. Source data are provided with this paper. All protein sequence database used in this study, including Human-reviewed, Glycoprotein-reviewed, B-cell-specific, and Random-1000 are listed in Supplementary Data 1. All data including the names of the.raw files and corresponding figures are provided with this paper (see Supplementary data 7).

## Code availability

The developed GlycoBinder is available on GitHub (https://github.com/IvanSilbern/GlycoBinder).

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

## Acknowledgements

We thank Wenfeng Zeng from the Key Lab of Intelligent Information Processing of Chinese Academy of Sciences (CAS), Institute of Computing Technology, Beijing, China, for software support for pGlyco 2. P.F. was supported by a Manfred-Eigen-Fellowship from the Max Planck Institute for Biophysical Chemistry. H.U. is funded by a Collaborative Research Center of the Deutsche Forschungsgemeinschaft (SFB1286).

## Author contributions

K.T.P. and H.U. planned the study. P.F. and Y.J. conceived the study. I.S. conceived the data processing software. C.D. and T.O. contributed to biological experiments. M.N. contributed to data analysis. C.L. contributed to the manuscript. K.T.P. and H.U. supervised the project, P.F., Y.J. and K.T.P. wrote the manuscript with help from all authors.

## Funding

## Competing interests

The authors declare no competing interests.
