## [Peer Review File · Nature Communications]

REVIEWER COMMENTS

Reviewer #1 (Remarks to the Author):

This paper's major claim is a new and potentially advantageous method for quantitative glycopeptide analysis called SugarQuant. SugarQuant builds on the SPS MS3 method pioneered by Gygi, which has previously been used for ordinary peptides, but not to my knowledge for glycopeptides. Along with methods development, the paper reports on GlycoBinder, a software tool that includes TMT quantitation, along with code for merging MS2 and MS3 scans for identification by pGlyco, an existing glycopeptide identification tool. The paper relies on previous methods and tools (SPS MS3 and pGlyco), but it also represents some clever original thinking along with a large amount of laboratory work: parameter optimization, data acquisition, and software development.

The paper is convincing and it should be of interest to almost all researchers working in glycoproteomics. SPS MS3 actually makes more sense for glycopeptides than for ordinary peptides, because Y0, Y1, etc. are sure to include the N-terminal TMT tag, whereas a y-ion from a peptide ending in something other than K will not be tagged. SPS MS3 also has the benefit of improving fragmentation coverage, because Y0, Y1, etc. often produce better peptide backbone fragmentation than does the full glycopeptide. The lymphoma cell experiment is an impressive demonstration of the method.

Even though most glycoproteomics researchers will be interested, not that many of them will want to adopt the method. First, the method virtually requires a high-end Orbitrap instrument capable of MS3. Second, TMT reagents are expensive. And most importantly, there are some significant tradeoffs in the method: it would not be advantageous when data acquisition speed is important, when glycopeptides are only a small part of the overall sample, or in simple samples when precursor mass and (reverse-phase) elution time, along with a little bit of MS2 information, is sufficient for glycopeptide identification. The paper would be improved by more quantitative discussion of the tradeoffs. For example, I would like to know how the cycle time compares to other Orbitrap "tribrid" data acquisition methods, such as triggered ETD, and also how the quantitation compares to TMT without SPS MS3 and to label-free quant by XIC integration.

The revised paper should also include a little more discussion of the scope and applicability of the methods. Can GlycoBinder software also be used with other glycopeptide software, for example, Byonic or BiopharmaFinder? Can the SugarQuant approach be extended to peptides with O-linked glycosylation?

One issue not addressed by the paper is in-source dissociation (ISD) of glycopeptides; this can be a real problem for quantitation. The revised version of the paper should at least

say something about this problem. Can GlycoBinder help with this problem by merging ISD scans into the "parent" scan?

Specific Small Comments:

Page 2, line 86, you mean "at most two fractions" not "less than two fractions".

Typos or misspellings: "Labled" and "Unlabled" in Supp. Fig 1f.

Page 4, lines 151 - 152, and also Figure 2(c). How important is pGlyco's glycan score relative to peptide score? If you have good Y0 and Y1 peaks, or alternatively both b- and y-ions for the peptide, do you really need to see more than one or two oxonium ions? Is the glycan score based on numbers of peaks matched or does it take into account reliability of peaks? Is it possible to obtain a high glycan score based on common ions such as 168, 186, 204, etc., even if expected ions (such as 274 for NeuAc or 290 for NeuGc) are missing?

Figure 3(c) -- the gray dots all blur together. Is it possible to make them smaller?

There should be a guide to reading the names of the .raw files. I imagine that this data will be made public when the paper is published?

Reviewer: Dr. Marshall Bern

Reviewer #2 (Remarks to the Author):

There is clearly a pressing need for improved tools for detection, characterisation, and especially quantitation of protein glycosylation in glycoproteomic workflows. This manuscript by Fang et al makes a substantial and novel contribution towards these ends. The workflow combines Glyco-SPS-MS3 for improved glycopeptide sample preparation, enrichment, and detection, with GlycoBinder for identification and data-processing.

The workflow improves TMT-labelled glycopeptide quantification by inclusion of MS3 workflows with automated selection of Y ions generated by MS2 HCD. This is a powerful novel workflow for quantitative glycoproteomics, increasing not only glycopeptide identification, but also reliable quantification. While the results describe increased identification of glycopeptides, more detail of how these identifications are validated and/or confirmed should be included. In particular, the details of the "FDR control on both peptide and glycan levels" are not transparent or described in detail.

In addition to providing improved glycopeptide identification and quantification, the Glyco-SPS-MS3 approach improves co-isolation interference. This is a valuable analytical improvement, that is well supported by the results.

The limitations introduced by the extended duty cycle time are acknowledged, and minimised by method optimisation.

The figures in both the main text and supplementary information are very high level summaries of processed data. Illustrative raw spectra should be presented to demonstrate the effectiveness of the Glyco-SPS-MS3 workflows in increasing the information content of glycopeptide fragmentation spectra. (e.g. annotated standard HCD MS2 spectra vs MS3 spectra from the Glyco-SPS-MS3 workflow for selected glycopeptides).

The workflows are used in a proof-of-principal experiment testing the effect of 2FF inhibition of fucosylation in Burkett's lymphoma cells. This is an impressive large-scale glycoproteomic characterisation dataset of these cells, and the 2FF inhibition is also an interesting experiment. Some intriguing links between fucosylation of B-cell receptor effectors and signalling are postulated.

It is a little peculiar that an improvement in performance with Glyco-SPS-MS3 is only seen at higher concentrations of 2FF. The authors rationalise this as particular difficulties in measuring non-fucosylated versus fucosylated glycopeptides with MS2. The logic of this explanation is not completely clear, and should be clarified, as well as the lack of performance improvement at low 2FF concentrations. No technical or biological validation of these results is included. Selected results of particular biological or technical interest should be appropriately validated to confirm the utility of the workflow.

The statistical analysis is clear and appropriate. Methods are clearly described, and bioinformatic workflows are available.

Reviewer #3 (Remarks to the Author):

In this study, an integrated glycoproteomic workflow, SugarQuant comprising cell lysis, protein extraction, protein concentration and protease digestion using protein aggregation capture, TMT labeling, N-glycopeptide enrichment, basic reverse phase fractionation, LC-MS analysis using Glyco-SPS-MS3, data processing using a novel GlycoBinder tool for MS2 and MS3 based identification and quantification. The SugarQuant workflow was applied to the analysis of Burkitt's lymphoma cells treated with varying doses of 2-deoxy-2-fluoro-L-fucose as a proof of concept study and demonstrated that the reduced N-fucosylation following treatment. The workflow combined several existing methods as well as a newly developed GlycoBinder tool for glycoproteomic analysis. The study optimized each step of the workflow. This manuscript fits in Nature Communications' high impact article. However, the study lacks of description or data for several key components for the platform evaluation.

1. How does the platform evaluate the false positive rate for glycopeptide identification and quantification?
2. Reproducibility is critical component of the platform analytical performance, which should be included in the evaluation in overall performance.
3. How does the combined platform outperform the existing platform?
4. The study only identified glycan composition, therefore, the glycan structure presentation should be avoid unless glycan structures were identified.
5. The supplementary tables need additional description or data organization.

REVIEWER COMMENTS

Reviewer #1 (Remarks to the Author):

This paper's major claim is a new and potentially advantageous method for quantitative glycopeptide analysis called SugarQuant. SugarQuant builds on the SPS MS3 method pioneered by Gygi, which has previously been used for ordinary peptides, but not to my knowledge for glycopeptides. Along with methods development, the paper reports on GlycoBinder, a software tool that includes TMT quantitation, along with code for merging MS2 and MS3 scans for identification by pGlyco, an existing glycopeptide identification tool. The paper relies on previous methods and tools (SPS MS3 and pGlyco), but it also represents some clever original thinking along with a large amount of laboratory work: parameter optimization, data acquisition, and software development.

The paper is convincing and it should be of interest to almost all researchers working in glycoproteomics. SPS MS3 actually makes more sense for glycopeptides than for ordinary peptides, because Y0, Y1, etc. are sure to include the N-terminal TMT tag, whereas a y-ion from a peptide ending in something other than K will not be tagged. SPS MS3 also has the benefit of improving fragmentation coverage, because Y0, Y1, etc. often produce better peptide backbone fragmentation than does the full glycopeptide. The lymphoma cell experiment is an impressive demonstration of the method.

Reply: We thank Dr. Marshall Bern for his thorough and insightful evaluation of our manuscript. We have further emphasized the advantages of Glyco-SPS-MS3 in the main text (page 4, second paragraph). We also provide representative spectra of Glyco-SPS-MS3 in the supplementary figure 5 for better visualization of the benefit of merged MS2 and MS3 scans for glycopeptide identification as compared to the standard MS2 scans.

Even though most glycoproteomics researchers will be interested, not that many of them will want to adopt the method. First, the method virtually requires a high-end Orbitrap instrument capable of MS3. Second, TMT reagents are expensive.

Reply: Our results demonstrate the advantages of Glyco-SPS-MS3 for improved glycopeptide identification and quantification. Particularly, Glyco-SPS-MS3 minimizes the co-isolation interference and the resulting ratio compression that frequently hampers standard DDA MS/MS analyses of chemically labeled glycopeptides. We are convinced that Glyco-SPS-MS3 is the method of choice for quantitative glycoproteomics. However, we agree with Dr. Bern that, like common SPS-MS3 methods, Glyco-SPS-MS3 requires a high-resolution mass spectrometer capable of MS3, which is not always affordable or accessible to all labs. We thus allow the GlycoBinder to process common MS2 raw files so that the researchers can still benefit from the automatic data analysis pipeline as well as the 2-step database search. Please see supplementary note 4 and our GitHub site (<https://github.com/IvanSilbern/GlycoBinder>) for more details.

We also understand the high cost of isobaric labeling reagents can limit their adoption, especially when large sample amounts are required. To alleviate it, we evaluated the use of reduced amounts of TMT reagents, down to a TMT-to-peptide ratio (wt/wt) of 2:1, which represents a four-fold reduction in required TMT-reagent amounts compared to the manufacturer's recommendations (800 μ g TMT to 100 μ g peptides, 8:1). Even at these reduced levels, we achieved over 99% of labeling efficiency. For every biological replicate of the fucosylation-inhibited samples, we labeled 400 μ g peptides in each condition using only one set of TMT reagents (800 μ g each). After combining, a total of 2.4 mg labeled peptides were obtained for the following sample preparation steps. We have added these discussions to supplementary note 2.

And most importantly, there are some significant trade-offs in the method: it would not be advantageous when data acquisition speed is important, when glycopeptides are only a small part of the overall sample, or in simple samples when precursor mass and (reverse-phase) elution time, along

with a little bit of MS2 information, is sufficient for glycopeptide identification. The paper would be improved by more quantitative discussion of the tradeoffs.

Reply: We agree with Dr. Bern that acquisition speed is a significant trade-off of the Glyco-SPS-MS3 and have thoroughly examined key MS parameters to optimize the method. We have carefully discussed the limitations of Glyco-SPS-MS3 in the main text (page 4, third paragraph) and in supplementary note 2. Specifically, we evaluated the injection time and AGC target using digested peptides of purified IgM. The IgM peptides were not enriched for glycopeptides, and only 15% of the resulting MS/MS scans contained glycan oxonium ions. Our results showed that prolonged cycle time (higher AGC targets and longer injection time) in MS2 helped to identify more glycopeptides with either pGlyco or Byonic search engines (Supp. Fig. 7). We reason that higher AGC and injection time benefit the identification of low abundance and/or low ionization efficiency glycopeptides in simple samples. Our results reveal that Glyco-SPS-MS3 can be beneficial for analyzing simple sample (one purified glycoprotein). For more complex samples, conventional MS2 methods still offer faster duty cycle, but SugarQuant showed comparable performance in terms of number of identified and quantified glycopeptides (supplementary figure 12).

We also agree that it becomes more challenging when dealing with samples that contain only a small portion of glycopeptides. We thus confirmed that the enrichment efficiency of ZIC-HILIC was not affected by TMT-labeling (supplementary figure 1f). In our opinion, a proper enrichment of glycopeptides is more important than raw MS acquisition speed under this circumstance, as non-glycopeptides often suppress the ionization of glycopeptides.

For example, I would like to know how the cycle time compares to other Orbitrap "tribrid" data acquisition methods, such as triggered ETD, and also how the quantitation compares to TMT without SPS MS3 and to label-free quant by XIC integration.

Reply: We thank Dr. Bern for making this point. We have discussed the benefits and drawbacks of different quantification approaches, including label free, SILAC, and chemical isobaric labeling, for glycopeptides in supplementary note 1. We have also compared the performance of our Glyco-SPS-MS3 with common MS2 methods in the manuscript (page 4, third paragraph, page 8, third paragraph, and supplementary note 3). We have now included extended discussion about the comparison of cycle time between Glyco-SPS-MS3 and other existing methods in supplementary note 3 (last paragraph) as follows:

“We compared the required cycle time of our Glyco-SPS-MS3 with previously published acquisition methods including HCD triggered AI ETD, ETD and EThcD methods by Rieley *et al*¹. The MS1 acquisition parameters were essentially the same, requiring 120,000 resolution at 200 m/z and maximum injection time of 50 ms and automatic gain control set to 400,000 (Rieley *et al*) or 500,000 (SugarQuant). The cycle time was automatically controlled by the machine and set to 3 s. Therefore, the main factor that limits acquisition speed and affects the overall cycle time is the total maximum injection time allowed for the MS2-MS3 or HCD-triggered (AI)ET(hc)D scan pairs. Riley *et al*. allocated 460 ms for an HCD/triggered (AI)ET(hc)D scan pair, including 60 ms for a survey HCD scan and 400 ms for a triggered (AI)ET(hc)D scan. In our suggested Glyco-SPS-MS3 settings, we allocated 500 ms for an MS2/SPS MS3 scan pair in total, with 150 ms being used by an MS2 scan and 350 ms being allocated for an SPS MS3 scan. This resulted in an 8% increase in the total time required for an MS2/SPS MS3 scan pair as compared with the triggered ETD methods. The required ion reaction time for (AI-)ET(hc)D, however, is often longer than that for HCD, which makes the difference in overall cycle time between the methods negligible. Importantly, SugarQuant utilizes both MS2 and SPS-MS3 scans and thus brings in additional advantages for reliable glycopeptide identification as well as accurate quantification.”

The revised paper should also include a little more discussion of the scope and applicability of the methods. Can GlycoBinder software also be used with other glycopeptide software, for example, Byonic or BiopharmaFinder?

Reply: Indeed, it is feasible to incorporate other glycopeptide identification software, such as Byonic, to SugarQuant. Currently, pGlyco 2.0 is the only search engine that is fully integrated in the GlycoBinder workflow. Nonetheless, we would also like to encourage readers to use other glycopeptide search engines. For that, GlycoBinder reports merged MS2/MS3 spectra in mgf format that are located in “pparse_output” folder and are marked with the “_mod.mgf” suffix. These mgf files can be used with any other search engine supporting mgf input. However, the resulting identification lists (i.e. all identified GPSMs) need to be integrated with the quantitative data from RawTools output (“_Matrix.txt” files in “rawtools_output” folder) manually. Scan numbers and raw file names can be used to integrate qualitative and quantitative information, respectively. One can also obtain the multi-dimension quantification by propagating the intensities of GPSMs to glycoforms, glycosites and glycans. We added a paragraph “Special case: use of another search engine” to Supplementary note 4 to highlight this particular use case. We have tested the feasibility of this manual strategy using Byonic.

We have also introduced an alternative way to make use of the Glyco-SPS-MS3 results via ProteomeDiscoverer (PD) in the online methods (page 15, fourth paragraph):

“For processing Glyco-SPS-MS3 results using PD, the nodes of “Spectrum selector” and “Spectrum grouper” were used for converting and combining spectra from MS2 and MS3, respectively, with the following settings. Precursor mass criterion: same singly charged mass; Precursor mass tolerance: 0.1 ppm; Max. RT difference: 0.04 min; Allow mass analyzer mismatch: False; Allow MS order mismatch: True. The resulting mgf files were used for pGlyco 2 searches. Alternatively, the mgf files can be used for glycopeptide identification and quantification using Byonic node in the PD platform.”

We still recommend to use GlycoBinder for automatic data analysis. Because the output formats of different search engines are quite different from the pGlyco, it is difficult for us to integrate multiple search engines to GlycoBinder.

Can the SugarQuant approach be extended to peptides with O-linked glycosylation?

Reply: SugarQuant in principle should work for O-glycopeptide quantification, but more systematic evaluation and optimization are required for good performance. Firstly, the PAC-based sample preparation can be used for extraction, digestion, and TMT labeling of proteins from a variety of biological samples. However, the efficiency of enriching O-glycopeptide by ZIC-HILIC need to be further optimized, or other means of enrichment methods have to be used. Secondly, the MS acquisition method should also be modified for O-glycopeptides. For example, mucin type O-glycosites occur most frequently in long serine/threonine rich sequences. HCD-centric methods as used in the Glyco-SPS-MS3 workflow may not able to confidently localize O-glycosites under these circumstances. We will evaluate all necessary improvements to make SugarQuant suitable for quantitative O-glycoproteomics.

One issue not addressed by the paper is in-source dissociation (ISD) of glycopeptides; this can be a real problem for quantitation. The revised version of the paper should at least say something about this problem. Can GlycoBinder help with this problem by merging ISD scans into the "parent" scan?

Reply: We appreciate the question. In-source dissociation (ISD) is indeed problematic for accurate quantification of glycopeptides, especially for those bearing sialic acid. When a glycopeptide is fragmented and loses a part of its glycan moiety in the source, the resulting fragmented pseudo-glycopeptide and its original “parent” glycopeptide will appear at the same retention time in the liquid chromatography. Once the chromatography does not separate glycopeptides sharing the same peptide sequence but bearing different glycans apart, the ISD glycopeptide may result in overlapping ion

chromatograms with its natural, unfragmented isomers. MS1-based quantification, including label-free and SILAC, will be compromised by both the reduced intensity of fragmented glycopeptide and the overlapped ion chromatogram. Isobaric labeling-based quantification, on the other hand, is not affected by the former because glycopeptides from all samples are pooled and should theoretically experience equal effects of ISD. However, the ISD glycopeptide and its natural isomers can be co-selected for MS2 and MS3 analyses, leading to an inaccurate quantification of natural glycopeptides. Unfortunately, this effect cannot be eliminated by simply merging the ISD scans into the “parent” scans. We are limited by the current technology and have no perfect solution for it.

We have added this discussion to the supplementary note 1.

Specific Small Comments:

Page 2, line 86, you mean "at most two fractions" not "less than two fractions".

Typos or misspellings: "Labeled" and "Unlabeled" in Supp. Fig 1f.

Reply: We apologize for the mistakes. We have corrected them.

Page 4, lines 151 - 152, and also Figure 2(c). How important is pGlyco's glycan score relative to peptide score? If you have good Y0 and Y1 peaks, or alternatively both b- and y-ions for the peptide, do you really need to see more than one or two oxonium ions? Is the glycan score based on numbers of peaks matched or does it take into account reliability of peaks? Is it possible to obtain a high glycan score based on common ions such as 168, 186, 204, etc., even if expected ions (such as 274 for NeuAc or 290 for NeuGc) are missing?

Reply: pGlyco 2 calculates the glycan scores based on the numbers of matched peaks, mass errors of matched ions, and the number of matched trimannosyl core ions. It then estimates the glycan FDR using a glycan decoy method coupled with a finite mixture model algorithm. For more detailed explanation of pGlyco 2 algorithm, please see **Point 1** below and its original publication². To our knowledge, pGlyco does not consider oxonium ions except NeuAc and NeuGc for glycan scoring, but uses them to determine whether the selected precursor is a glycopeptide. Glycan score and glycan FDR are important to filter out GPSMs that fail to inform the glycan composition and to differentiate isobaric glycopeptides with the same peptide sequence but differed glycan composition.

Nevertheless, peptide score and peptide FDR still have major effects in GPSMs filtering in pGlyco2. In our experience, peptide FDR can filter out ~60-80% low-scoring GPSMs, while glycan FDR alone removed only about 40% GPSMs.

Figure 3(c) -- the gray dots all blur together. Is it possible to make them smaller?

Reply: We used Excel to generate this figure and have already selected the smallest available dot symbols. Due to the high degree of correlation between the intensities of reporter ions from RawTools and PD, the dots align perfectly on the trend line, which may cause the blurred appearance depending on graphical resolution of the display.

There should be a guide to reading the names of the .raw files. I imagine that this data will be made public when the paper is published?

Reply: We do have provided a table (supplementary data 7_raw data list) in the supplementary materials of our original manuscript summarizing names of all raw data and their corresponding figures. We regret that it may not be clear enough and have noted this table in the “Data Availability”. We have uploaded all raw files to the ProteomeXchange Consortium via the PRIDE partner repository with the dataset identifier PXD018349. The reviewer account has been in the cover letter to the editor:

Username: reviewer98779@ebi.ac.uk

Password: MrYJdqLm

The raw data will be made public as soon as the manuscript is accepted.

Reviewer: Dr. Marshall Bern

Reviewer #2 (Remarks to the Author):

There is clearly a pressing need for improved tools for detection, characterisation, and especially quantitation of protein glycosylation in glycoproteomic workflows. This manuscript by Fang et al makes a substantial and novel contribution towards these ends. The workflow combines Glyco-SPS-MS3 for improved glycopeptide sample preparation, enrichment, and detection, with GlycoBinder for identification and data-processing. The workflow improves TMT-labelled glycopeptide quantification by inclusion of MS3 workflows with automated selection of Y ions generated by MS2 HCD. This is a powerful novel workflow for quantitative glycoproteomics, increasing not only glycopeptide identification, but also reliable quantification.

While the results describe increased identification of glycopeptides, more detail of how these identifications are validated and/or confirmed should be included. In particular, the details of the "FDR control on both peptide and glycan levels" are not transparent or described in detail.

Reply: We are delighted that the reviewer appreciate our work and considers it as an important and timely study. SugarQuant relies on the previously published glycopeptide database search engine, pGlyco 2, for confident glycopeptide identification. pGlyco 2 scores the peptide sequence and glycan composition separately based on matched peptide b-/y-ions and on matched glycan Y ions, respectively. Peptide FDR is then calculated using target-decoy approach and glycan FDR is estimated using the glycan decoy method coupled with a finite mixture model algorithm. For more detailed explanation of pGlyco 2 algorithm, please see **Point 1** below and its original publication².

In addition to providing improved glycopeptide identification and quantification, the Glyco-SPS-MS3 approach improves co-isolation interference. This is a valuable analytical improvement, that is well supported by the results. The limitations introduced by the extended duty cycle time are acknowledged, and minimised by method optimisation. The figures in both the main text and supplementary information are very high level summaries of processed data.

Illustrative raw spectra should be presented to demonstrate the effectiveness of the Glyco-SPS-MS3 workflows in increasing the information content of glycopeptide fragmentation spectra. (e.g. annotated standard HCD MS2 spectra vs MS3 spectra from the Glyco-SPS-MS3 workflow for selected glycopeptides).

Reply. We thank the reviewer for the suggestion. We have manually annotated MS2 and MS3 spectra from a Glyco-SPS-MS3 analysis of a TMT-labeled IgM peptide (m/z 1238.89³⁺, YK Δ NSDISSTR+5Hex5HexNAc1Fuc, N is N-glycosylation site, TMT 6 modifications on N-terminal and K, supplementary figure 5). We also copy the figure below. We also compared the merged spectrum (MS2+MS3) with a standard HCD MS2 spectrum from the same glycopeptide.

b Merged MS₂ and MS₃

c Standard MS₂

Supplementary figure 5. Representative MS₂ and MS₃ spectra of a glycopeptide YKNSDISSTR+Hex5HexNAc5Fuc acquired using standard HCD MS₂ or Glyco-SPS-MS₃. (a) Annotated MS₂ (upper) and MS₃ (bottom) spectra of the glycopeptide using Glyco-SPS-MS₃. All matched Y ions (red lines), b/y ions (yellow lines), and reporter ions (green ions) are marked. The top ten precursors selected for MS₃ are marked with blue lines. Magnified *m/z* regions of the spectra are highlighted with the magnitude specified. (b) The MS₂ and MS₃ spectra from (a) were merged and

subjected to pGlyco 2 database search. The matched fragment ions were reported using pLabel, a spectral visualization tool bundled with pGlyco 2. (c) A standard HCD MS2 spectrum from the same selected glycopeptide.

The workflows are used in a proof-of-principal experiment testing the effect of 2FF inhibition of fucosylation in Burkett's lymphoma cells. This is an impressive large-scale glycoproteomic characterisation dataset of these cells, and the 2FF inhibition is also an interesting experiment. Some intriguing links between fucosylation of B-cell receptor effectors and signalling are postulated.

It is a little peculiar that an improvement in performance with Glyco-SPS-MS3 is only seen at higher concentrations of 2FF. The authors rationalise this as particular difficulties in measuring non-fucosylated versus fucosylated glycopeptides with MS2. The logic of this explanation is not completely clear, and should be clarified, as well as the lack of performance improvement at low 2FF concentrations.

Reply: We apologize for not having made this clear enough in the manuscript. We intended to emphasize that Glyco-SPS-MS3 readily determined the significantly decreased fucosylation in DG75 cells treated with higher doses of 2FF, which would have been missed by common MS2-only. In addition, Glyco-SPS-MS3 also detected minor changes in fucosylation even under lower concentration of 2FF treatments (again missed by MS2-only analysis). We have modified the sentence in the main text, and have prepared a new supplementary figure 13. We also copy the figure below.

The improvement of quantitative accuracy using Glyco-SPS-MS3 is a result of two factors: 1. the reduced co-isolation interference of TMT reporter ions from non-fucosylated glycopeptides in the quantitation of fucosylated glycopeptides; 2. significantly enhanced TMT reporter ion intensities using the Glyco-SPS-MS3 method (Supp. Fig. 11c).

Supplementary figure 13. Glyco-SPS-MS3 determined modestly decreased fucosylation in DG75 cells treated with lower concentrations of 2FF. Ratio distributions of glycoforms quantified via the Glyco-

SPS-MS3 and the MS2 method upon the treatment with lower 2FF concentrations (60 μ M, 120 μ M and 240 μ M) were aligned. Different 2FF concentrations were color-coded as shown in the figure.

No technical or biological validation of these results is included. Selected results of particular biological or technical interest should be appropriately validated to confirm the utility of the workflow. The statistical analysis is clear and appropriate. Methods are clearly described, and bioinformatic workflows are available.

Reply: We thank the reviewer for his/her suggestion. We did, in fact, perform an independent technical validation of our findings by blotting against a biotinylated fucose-specific lectin (*Aleuria Aurantia* lectin, AAL). In agreement with the mass spectrometric results, the AAL blotting showed dose-dependent reduction of fucosylation (Supplementary figure 12). In addition, we added information on the reproducibility of mass spectrometric replicates to demonstrate the stability of our workflow (see below). We chose not to include functional biological validation experiments at this point, however, as this would have been outside of the scope of this study which focuses on the presented analytical workflow.

To confirm the reproducibility of SugarQuant, we performed the 2FF treatments in two biological replicates, and each biological replicate included technical triplicates. We examined the detected intensities of commonly identified glycoforms between replicates and showed good Pearson correlations (0.59-0.98) in all replicates. Samples treated with lower-dose 2FF (60 μ M and 120 μ M) showed relatively reduced correlation in replicates. Because 2FF only induced slight changes in glycoform quantities in those samples, the variation in quantitation may have been exaggerated in the Pearson correlation analysis. We have modified the main text accordingly and added supplementary figure 14 to disclose the reproducibility of SugarQuant pipeline. We copy the figure below.

Supplementary figure 14. An overview of Pearson correlations between replicates. Each biological replicate contains technical triplicates. The first biological replicate is marked in blue, and the second in green.

Reviewer #3 (Remarks to the Author):

In this study, an integrated glycoproteomic workflow, SugarQuant comprising cell lysis, protein extraction, protein concentration and protease digestion using protein aggregation capture, TMT labeling, N-glycopeptide enrichment, basic reverse phase fractionation, LC-MS analysis using Glyco-SPS-MS3, data processing using a novel GlycoBinder tool for MS2 and MS3 based identification and quantification. The SugarQuant workflow was applied to the analysis of Burkitt's lymphoma cells treated with varying doses of 2-deoxy-2-fluoro-L-fucose as a proof of concept study and demonstrated that the reduced N-fucosylation following treatment. The workflow combined several existing methods as well as a newly developed GlycoBinder tool for glycoproteomic analysis. The study optimized each step of the workflow. This manuscript fits in Nature Communications' high impact article. However, the study lacks of description or data for several key components for the platform evaluation.

Reply: We sincerely appreciate the reviewer's recognition of the quality of our work.

How does the platform evaluate the false positive rate for glycopeptide identification and quantification?

Reply: SugarQuant relies on the previously published glycopeptide database search engine, pGlyco 2, for confident glycopeptide identification. pGlyco 2 scores the peptide sequence and glycan composition separately based on matched peptide b-/y-ions and on matched glycan Y ions, respectively. Peptide FDR is then calculated using the target-decoy approach and glycan FDR is estimated using their glycan decoy method coupled with a finite mixture model algorithm. For more detailed explanation of pGlyco 2 algorithm, please see **Point 1** below and its original publication².

Reproducibility is critical component of the platform analytical performance, which should be included in the evaluation in overall performance.

Reply: We thank the reviewer for pointing this out. To confirm the reproducibility of SugarQuant, we performed the 2FF treatments in two biological replicates, and each biological replicate included technical triplicates. We examined the detected intensities of commonly identified glycoforms between replicates and showed good Pearson correlations (0.59-0.98) in all replicates. Samples treated with lower-dose 2FF (60 μ M and 120 μ M) showed relatively reduced correlation in replicates. Because 2FF only induced slight changes in glycoform quantities in those samples, the variation in quantitation may have been exaggerated in the Pearson correlation analysis. We have modified the main text accordingly and added supplementary figure 14 to disclose the reproducibility of the SugarQuant pipeline. The figure is copied above, as a reply to the comments of the 2nd reviewer.

How does the combined platform outperform the existing platform?

Reply: SugarQuant integrates fast PAC-based sample preparation, optimized Glyco-SPS-MS3 and the semi-automatic GlycoBinder script. It allows for confident, global identification and quantification of intact glycopeptides in complex biological samples. To the best of our knowledge, this is the first study evaluating and optimizing the entire glycoproteomics workflow, from sample preparation, MS acquisition, database search, and data analysis. Therefore, the overall improvement is multi-faceted.

Regarding sample preparation, the previously published NGAG (solid phase extraction of N-linked glycans and glycosite-containing peptides)³ applied a chemoenzymatic method to first identify glycosylation sites and then to determine the glycan heterogeneity on the sites. Despite its capability of comprehensive characterization of glycoproteins, the NGAG workflow involves multiple chemical reactions and is relatively time-consuming and labor-intensive. Besides, it may not be feasible to

further split a sample with a limited amount for separate de-glycopeptide and intact glycopeptide analyses. In contrast, Stadlmann *et al.*⁴ used urea and HILIC respectively for protein extraction and glycopeptides enrichment. This workflow is relatively simple but any residual urea can negatively impact TMT labelling efficiency. Our SugarQuant utilizes a PAC-based method to ensure sufficient solubilization of membrane-associated glycoproteins, complete and fast removal of detergents, efficient TMT-labeling, and selective glycopeptide enrichment with reduced handling time and sample loss. It is easy to handle even for non-expert researchers, especially with regard to sample preparation.

None of abovementioned studies evaluated the MS acquisition strategies, and they all developed their own database search algorithms. We developed the novel Glyco-SPS-MS3 strategy and demonstrate its performance in both glycopeptide identification and quantification. We integrated well-tested and freely available software tools in GlycoBinder for data processing and database search. In terms of identification, NGAG assigned 4,562 oxonium ion-containing spectra to 1,562 unique glycopeptides (containing 518 glycosites and 81 glycans) in OVCAR-3 cells using GPQuest software with filtering based on the presence of peptide+HexNAc and/or peptide ions, as well as ≥ 7 observed b and y ions (1% FDR). Stadlmann *et al.* developed SugarQB and identified 1,100 glycopeptides mapping to 576 proteins in human embryonic stem cells with a Mascot-based peptide FDR cut-off. SugarQuant identified and quantified over 5000 glycoforms containing 855 glycosites from 528 glycoproteins in DG75 cells using the pGlyco2 search engine with FDR control at three levels of matched glycan, peptide, and glycopeptide (for detail please see below **Point 1**).

All in all, we demonstrate that SugarQuant outperforms the existing platforms, and anticipate that SugarQuant will have broad applicability for biological and biomedicine research.

The study only identified glycan composition, therefore, the glycan structure presentation should be avoid unless glycan structures were identified.

Reply: We fully agree with the reviewer on this point. We have corrected the manuscript accordingly.

The supplementary tables need additional description or data organization.

Reply: We thank the reviewer for his/her suggestions. We have modified the titles of supplementary data and added a description sheet for each table. Supplementary data 7 summarizes all .raw files and their corresponding figures.

Point 1 (to all reviewers): pGlyco 2.0 for glycopeptide identification: scoring algorithm and FDR evaluation.

To better explain the scoring algorithm and FDR evaluation in pGlyco 2, we add below the details extracted from its original publication “pGlyco 2.0 enables precision N-glycoproteomics with comprehensive quality control and one-step mass spectrometry for intact glycopeptide identification”².

pGlyco 2.0 is an integrated search engine specifically designed for the interpretation of glycopeptide SCE-HCD-MS/MS spectra. The procedures of glycopeptide identification in pGlyco 2.0 includes coarse-scoring, fine-scoring and GPSM FDR analysis of glycopeptide.

The first step is coarse-scoring, which was an open search mode for the analysis of the glycan candidates. Given a spectrum, in the coarse-scoring step, for each glycan in the glycome database, the associated peptide backbone mass was calculated as the precursor mass of this spectrum minus the glycan mass, and then the associated masses of all Y ions (glycan fragment ions with peptide backbone attached) could be deduced. Each glycan was scored by the number of matched Y ions, and any glycan with less than 2 trimannosyl core ions matched will be filtered out. Moreover, the top-k (K = 100 by default) candidate glycans were kept for the fine-scoring step. The second step is fine-scoring, which was a scoring scheme for a glycopeptide-spectrum match (GPSM). For each valid

glycan candidate after coarse scoring, the corresponding candidate peptides were searched by pFind based only on the peptide backbone mass. After peptide search, the candidate glycopeptide candidates were generated by combining glycan candidates and peptide candidates, and fine-scoring was then performed for the GPSM to obtain scores for the glycan, peptide and glycopeptide. The scoring scheme of glycan considered the matched peaks, their matching mass errors and the number of matched trimannosyl core ions:

$$\text{Score}_G = \sum_i \log(\text{inten}_i) \left(1 - \left|\frac{\text{merr}_i}{\text{tol}_i}\right|^4\right) (\text{ratio}_{\text{ion}})^\alpha (\text{ratio}_{\text{core}})^\beta. \quad (1)$$

In pGlyco 2.0, we have also developed a similar scheme for the scoring of peptide backbone:

$$\text{Score}_P = \sum_i \log(\text{inten}_i) \left(1 - \left|\frac{\text{merr}_i}{\text{tol}_i}\right|^4\right) (\text{ratio}_{\text{ion}})^\gamma. \quad (2)$$

The meanings of terms are: $\text{ratio}_{\text{core}} = \# \text{matched trimannosyl core ions} / \# \text{theoretical trimannosyl core ions}$; $\text{ratio}_{\text{ion}} = \# \text{matched ions} / \# \text{theoretical ions}$; merr_i is the matching mass error of the i -th matched peak; tol_i is the mass tolerance of the i -th matched peak. The total score of the glycopeptide was the weighted sum of these two scores:

$$\text{Score}_{GP} = w \times \text{Score}_G + (1 - w) \times \text{Score}_P. \quad (3)$$

The four parameters, α and β of Score_G , γ of Score_P and w of Score_{GP} were fine-tuned as $\alpha = 0.56$, $\beta = 0.42$, $\gamma = 0.94$ and $w = 0.35$ by Ranking SVM based on the SCE-HCD-MS/MS spectra. The fine-tuning process of Score_G , was described as an example: for a well-designed fine-scoring scheme, a correct match of a spectrum would always ‘beat’ other incorrect matches and be ranked as top-1. The aim of fine-tuning the parameters of the scoring scheme was to rank as many correct matches onto top-1 as possible. The learning-to-rank model was very suitable for this scenario. For Score_G , it was not easy to fine-tune the parameters because the score was an exponential form of the parameters. Taking the logarithm of Score_G would get a linear form of the parameters, which became:

$$\begin{aligned} \log(\text{Score}_G) &= \log\left(\sum_i \log(\text{inten}_i) \left(1 - \left|\frac{\text{merr}_i}{\text{tol}_i}\right|^4\right)\right) + \alpha \times \log(\text{ratio}_{\text{ion}}) \\ &+ \beta \times \log(\text{ratio}_{\text{core}}). \end{aligned}$$

This linear form could be easily modeled by Ranking SVM, which is a very popular learning-to-rank algorithm for machine learning. With manual inspection, we could get the correct and incorrect GSMs, and then the Ranking SVM model could be trained on these benchmark GPSMs. After the coarse-scoring and fine-scoring, pGlyco 2.0 performed GSPM FDR analysis. To our knowledge, there is no widely accepted protocol for FDR analysis of glycopeptide identification in glycoproteomics yet. We carefully studied the false glycopeptide identification and derived a new mathematical model for the FDR analysis of the intact glycopeptide identification. For a GSM, an incorrect identification of either the glycan or the peptide was called a false identification, so the FDR of glycopeptide could be written:

$$\widehat{\text{FDR}}(x) = p(G = \text{false} \cup P = \text{false} | X \geq x). \quad (5)$$

Here, $G = \text{false}$ and $P = \text{false}$ refer to the false identification of the glycan and the peptide respectively, and x is the given score threshold. Since $p(G \cup P) = p(G) + p(P) - p(G \cap P)$, Eq. (5) could be rewritten as

$$\begin{aligned} \widehat{\text{FDR}}(x) &= p(G = \text{false} | X \geq x) + p(P = \text{false} | X \geq x) \\ &- p(G = \text{false} \cap P = \text{false} | X \geq x), \quad (6) \\ \widehat{\text{FDR}}(x) &= \widehat{\text{FDR}}_G(x) + \widehat{\text{FDR}}_P(x) - \widehat{\text{FDR}}_{G \cap P}(x). \end{aligned}$$

For $FDR_G(x)$, it could be estimated by using our previously reported glycan decoy method coupled with a finite mixture model algorithm, and the $FDR_P(x)$ of peptide could be estimated as $\#pep_decoy_glycan_target/\#both_target$, $FDR_{G \cap P}(x)$ could be estimated by $\#pep_decoy_glycan_decoy$ and $\#both_target$.

References

1. Riley, N.M., Hebert, A.S., Westphall, M.S. & Coon, J.J. Capturing site-specific heterogeneity with large-scale N-glycoproteome analysis. *Nat. Commun.* **10**, 1311 (2019).
2. Liu, M.Q. et al. pGlyco 2.0 enables precision N-glycoproteomics with comprehensive quality control and one-step mass spectrometry for intact glycopeptide identification. *Nat. Commun.* **8**, 438 (2017).
3. Sun, S. et al. Comprehensive analysis of protein glycosylation by solid-phase extraction of N-linked glycans and glycosite-containing peptides. *Nat. Biotechnol.* **34**, 84-88 (2016).
4. Stadlmann, J. et al. Comparative glycoproteomics of stem cells identifies new players in ricin toxicity *Nature* **549**, 538-542 (2017).

REVIEWERS' COMMENTS:

Reviewer #1 (Remarks to the Author):

The revised manuscript addresses the points raised in the earlier reviews. This paper demonstrates a novel and useful method for quantitative glycoproteomics.

-- Marshall Bern

Reviewer #2 (Remarks to the Author):

The authors have provided convincing and appropriate responses to my concerns and comments, and made suitable revisions and additions to the manuscript. The workflow, tools, and data represent an important contribution to quantitative glycoproteomics that will make a substantial impact on the field.

Reviewer #3 (Remarks to the Author):

The revised manuscript addressed my comments and is acceptable for publication.